



# Leads and ridges in Arctic sea ice from RGPS data and a new tracking algorithm

Nils Hutter[1], Lorenzo Zampieri[1], and Martin Losch[1]

[1]Alfred-Wegener-Institut, Helmholtz Zentrum für Polar- und Meeresforschung, Bremerhaven, Germany.

**Correspondence:** Nils Hutter (nils.hutter@awi.de)

**Abstract.** Leads and pressure ridges are dominant features of the Arctic sea ice cover. Not only do they affect heat loss and surface drag, but also provide insight into the underlying physics of sea ice deformation. Due to their elongated shape they are referred as Linear Kinematic Features (LKFs). This paper introduces two methods that detect and track LKFs in sea ice deformation data and establish an LKF data set for the entire observing period of the RADARSAT Geophysical Processor System (RGPS). Both algorithms are available as open-source code and applicable to any gridded sea-ice drift and deformation data. The LKF detection algorithm classifies pixels with higher deformation rates compared to the immediate environment as LKF pixels, divides the binary LKF map into small segments, and re-connects multiple segments into individual LKFs based their distance and orientation relative to each other. The tracking algorithm uses sea-ice drift information to estimate a first guess of LKF distribution and identifies tracked features by the degree of overlap between detected features and the first guess. An optimization of the parameters of both algorithms is presented, as well as an extensive evaluation of both algorithms against hand-picked features in a reference data set. An LKF data set is derived from RGPS deformation data for the years from 1996 to 2008 that enables a comprehensive description of LKFs. LKF densities and LKF intersection angles derived from this data set agree well with previous estimates. Further, a power-law distribution of LKF length, an exponential distribution of LKF lifetimes, and a strong link to atmospheric drivers, here Arctic cyclones, is derived from the data set. Both algorithms are applied to output of a numerical sea-ice model to compare the LKF intersection angles in a high-resolution Arctic sea-ice simulation with the LKF data set.

## 1 Introduction

The Arctic sea ice cover is an aggregation of ice floes of different size that changes continuously due to thermodynamic and dynamic processes. Thermodynamic processes slowly modify the shape of floes by freezing and melting, but rapid changes in floes shapes are caused by the deformation of the brittle ice. The drivers of these events are mainly wind, ocean currents, tides and interaction with coastal geometries.

When the ice cover breaks, leads form along floe boundaries as strips of open ocean. In such an opening of the ice cover there is strong upward heat flux from the warm ocean to the cold atmosphere, causing new ice formation and changes of the albedo. Colliding ice floes form pressure ridges and ice keels that change both the atmosphere-ice and the ice-ocean drag



coefficient. Both leads and pressure ridges are usually elongated features with lengths ranging from a few meters up to hundreds of kilometers.

Multiple studies used large amounts and and a great variety of satellite imagery of the Arctic ocean to describe the characteristics of deformation features and gain insight into the underlying physics. Lead densities were derived from MODIS images in the thermal infrared for cloud-free parts of the Arctic ocean (Willmes and Heinemann, 2016), from AMSR-E passive microwave brightness temperatures (Röhrs and Kaleschke, 2012), and from CryoSat-2 altimeter data (Wernecke and Kaleschke, 2015). Bröhan and Kaleschke (2014) extracted Pan-Arctic lead orientations from passive microwave data using a Hough transform. Kwok (2001) provided a qualitative description of these deformation features based on drift observations derived from SAR imagery, combining leads and pressure ridges under the term Linear Kinematic Features (LKFs) due to their dynamic nature. All these studies avoid the problem of detecting individual LKFs by applying statistics over continuous fields such as sea ice deformation or concentration. Miles and Barry (1998) presented a 5-year climatology of lead density and orientation based on manual detection in thermal- and visible-band imagery. Manual detection was also used to study the intersection angles of LKFs and its inferences on the rheology describing sea-ice deformation (Erlingsson, 1988; Walter and Overland, 1993).

All of these studies are limited either to specific information (density or orientation) or to a short time-series due to laborious manual detection. First attempts to automatically extract LKFs from satellite data were based on skeletons to describe LKFs (Banfield, 1992; Van Dyne and Tsatsoulis, 1993; Van Dyne et al., 1998), but Van Dyne et al. (1998) suggested "knowledge-based techniques" to further divide a skeleton into individual branches. This idea was picked up in an algorithm that automatically detects LKFs as objects in sea-ice deformation data (Linow and Dierking, 2017). Only 10 RGPS snapshots were analyzed in this way, but many more snapshots are necessary for a comprehensive description of LKFs. As the method of Linow and Dierking (2017) does not contain a tracking algorithm for LKFs, important temporal characteristics such as lifetimes cannot be derived from their detected LKFs, but only spatial statistics.

With increasing resolution of classical (viscous-plastic) sea ice models (Hutter et al., 2018) or with new rheological frameworks (e.g. Maxwell elasto-brittle, Rampal et al., 2016; Dansereau et al., 2016), sea-ice models start to resolve small-scale deformation with larger floes and leads. Typical measures for evaluating the modeled LKFs include scaling properties of sea-ice deformation (Hutter et al., 2018) or lead area density (Wang et al., 2016). An evaluation of these simulations based on individual features would be far more comprehensive and thorough and would help to improve model physics.

The objective of this study is to develop an open-source algorithm that automatically detects deformation features in regular gridded sea-ice deformation data and then tracks them using drift data. For this purpose, we present a modified version of the detection algorithm of Linow and Dierking (2017) and introduce an automatic tracking algorithm that takes into account the advection of deformation features with the overall sea-ice drift as well as growing and and shrinking features. Both algorithms are applied to the entire RADARSAT Geophysical Processor System (RGPS) drift and deformation data set to produce a multi-year LKF data-set that makes a comprehensive description of spatio-temporal characteristics of LKFs possible.



## 2  Data

One main requirement of the LKF detection algorithm presented in this study is that it should be applicable to both satellite observations and output of numerical sea-ice models. Thus, we use deformation data to detect LKFs rather than passive microwave data (Bröhan and Kaleschke, 2014) or thermal infrared imagery (Willmes and Heinemann, 2016), which is usually
not simulated in a sea ice model. Sea ice deformation, which is derived from sea-ice drift, can be observed by satellite, ship radar, and buoys and it is also simulated by numerical models.

### 2.1  Deformation data

In this study, we use deformation data provided by the RADARSAT Geophysical Processor System (RGPS). This data set is based on sea ice drift derived by tracking ice motion in SAR images. In each freezing season points are initialized on a
regular 10-km grid that are tracked over the winter until the on-set of the melting season. A Lagrangian deformation data set is computed from these trajectories using line integral approximations. Data is available for the years 1997 to 2008 with varying spatial coverage of the Amerasian Basin. We use the gridded version of the RGPS data-set for our analysis, which is interpolated on to a regular grid with 12.5 km grid spacing.

### 2.2  Drift data

To track features detected by the LKF detection algorithm in RGPS deformation data, the drift between two RGPS records is required for an a-priori guess of the temporal continuation of the individual feature. In the RGPS data set the derived deformation data is published along with the original drift data that is used for the deformation rate computation. Since RGPS drift is only provided as a Lagrangian data set, we interpolate the drift to the same regular 12.5-km grid on which the deformation data is provided.

### 20  2.3  Evaluation data-set

Automated object detection requires an evaluation against validation data. For this purpose, we use the data set of hand-picked LKFs presented in Linow and Dierking (2017). This data set comprises 1411 LKFs detected by human eye for 12 RGPS snapshots (December 29, 2005 to February 2, 2006). The intrinsic localization uncertainty of the visual detected features was shown to be 0.75px and the uncertainty in the line length 8% (Linow and Dierking, 2017).
Since this data set only provides LKFs for single snapshots but does not include information about the temporal evolution of LKFs between different snapshots, we need to expand the data set in this regard. In doing so, we advect the hand-picked LKFs of one snapshot using the RGPS drift to get an a priori guess of LKF position in the next snapshot. We visually compare the advected LKFs from the previous snapshot to LKFs of the next snapshot. If two LKFs overlap and agree in the entire overlapping area in position, shape, and orientation, they are marked as tracked LKF. Furthermore, each tracked LKF is described by
probability, degree of overlap, and type of shape change (no change, growing, shrinking, and branching), which are all visually estimated. In total 392 LKFs were tracked within these 12 RGPS snapshots, which corresponds to $\sim 28\%$ of the LKFs in the





evaluation data set. For the remaining 1019 LKFs no matching LKF in the next record is found. Thus, these LKFs have a lifetime that is shorter than the temporal resolution of 3 days if errors in the manual tracking are not considered.

## 3  LKF detection

### 3.1  Method description

Our LKF detection algorithm consists of three parts: (1) a preprocessing step that transforms the input deformation data into a binary map of pixels that mark LKFs by using different filter steps, (2) a detection routine that splits the network of LKF pixels in the binary map into the smallest possible segments, and (3) a reconnection instance that estimates the probability of different segments belonging to one feature and then connects all segments of a LKF. The general structure of the algorithm follows Linow and Dierking (2017), although individual parts have been modified substantially. The main enhancements of the
algorithm are a parallel detection of segments with a stronger constraint on the curvature and the introduction of a probability based reconnection. Further, the entire algorithm was rewritten in Python (Python Software Foundation, http://www.python. org) to avoid license issues with the previous code that was based on the commercial software IDL.

### 3.1.1  Data preprocessing and filtering

The standard input data of the LKF detection algorithm is the total deformation rate of sea ice $\dot{\epsilon}_{tot} = \sqrt{\dot{\epsilon}_I^2 + \dot{\epsilon}_{II}^2}$ that can be
derived from both satellite data and model output (Fig. 1a). LKFs are marked by regions of high deformation rates, because they are located along the boundaries of ice floes where most deformation takes place. The actual magnitude of deformation along an LKF, however, varies with the background deformation and the spatial scale, because of its multi-fractal properties (Weiss, 2013). Thus a simple thresholding of deformation rates is not sufficient to filter LKFs. Instead, we are interested in detecting deformation that is notably higher than the local environment. As LKFs are lines of high deformation, we need to
detect edges in the deformation field.

Prior to the edge detection, we take the natural logarithm of the input field (Fig. 1b) and perform a histogram equalization (Fig. 1c). Both highlight the local differences across different scales and enhance the contrast in regions of low deformation rates.

We use a Difference of Gaussian filter (DoG) following Linow and Dierking (2017) for the edge detection (Fig. 1d). The
DoG filter subtracts two filtered versions of the same input data: the first is smoothed with a Gaussian kernel of radius $r_1 = 1$ corresponding to a half-width $\sigma_1 = 0.5$, and the second is smoothed with a Gaussian kernel of radius $r_2 = 5$ (half-width $\sigma_2 = 2.5$), where $r_1 < r_2$. The smaller radius $r_1$ corresponds to the smallest scale of features that will be detected by the DoG and the second radius $r_2$ provides the upper limit of the scales detected. Note that this scale limitation applies to the width of the LKFs as well as to their lengths. For edges of scale $r_1 < L < r_2$, the DoG-filtered values are positive because the local
deformation rate is higher than in the environment of radius $r_2$. Pixels are marked as LKFs when the DoG-filtered pixels are



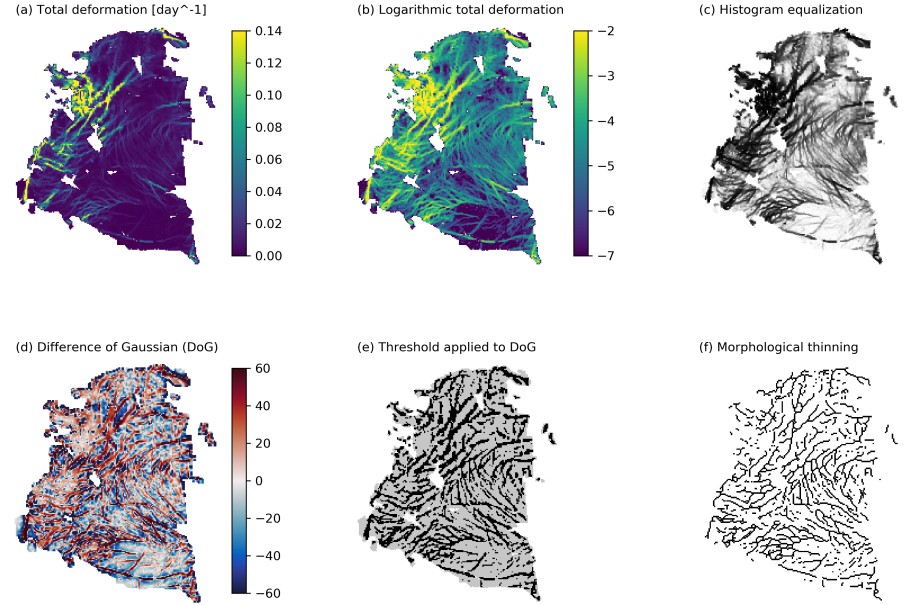

**Figure 1.** Filter sequence from (a) input field total deformation to (f) final output the morphological thinned binary map of LKF pixels. Intermediate steps are (b) logarithmic deformation, (c) histogram equalization, (d) Difference of Gaussian filter, and (e) the thresholded output of the DoG filter. RGPS deformation data for January 1st, 2006 is used for this example.

larger than a positive threshold $d_{\mathrm{LKF}}$. The result is a binary map where pixels with value 1 belong to LKFs (see Fig. 1e for a threshold of 15).

At this point, LKFs in the binary map are still represented in their original width. To detect which pixels belong to which LKF we add a further level of abstraction and reduce the width of all LKFs to one pixel (Fig. 1f). To this end, a morphological
thinning algorithm reduces the binary map to its skeleton. We use the `skeletonize` function of the open-source python package `scikit-image` (van der Walt et al., 2014) based on Zhang and Suen (1984). Skeletonization was used before to detect leads in original or classified, that is, preprocessed and charted, SAR images (Banfield, 1992; Van Dyne and Tsatsoulis, 1993; Van Dyne et al., 1998).

### 3.1.2 Segment detection

We detect small segments of pixels that contain parts of one LKFs in the binary map. Based on the morphological thinned binary map, groups of pixels that form a line are detected. In this first detection step, we want to guarantee that all pixels of a detected segment belong to the same LKF. Therefore, we detect the smallest segments possible that are in the simplest case the points in between intersections of lines in the binary map (Fig. 1f).

As a starting point of the segment detection we use LKF pixels that have only one neighbouring cell also marked as LKF.
Within each iteration, the detection algorithm proceeds to the LKF neighbour and checks again the number of neighbouring





cells for LKFs. If the new cell has also only one neighbouring LKF cell (neglecting the cell from the prior iteration) the search is continued. If the new cell has more than one neighbouring LKF cell, that is, it is a junction, the detection cycle is stopped and these neighbouring points become the new starting points. Besides the number of neighbouring LKF cells, a change in direction compared to the orientation of the last 5 pixels can also terminate the detection cycle: if the angle between the line

connecting the centres of the potential new cell and the current cell and the linear fit to the previous 5 pixels of the segment exceeds $45°$, the detection cycle is interrupted and the new cell is marked as a new starting point. If the segments is still shorter than 5 pixels, all available pixels are taken into account. We use 5 pixels in contrast to 2 pixels used by Linow and Dierking (2017) to impose a stronger constraint on the curvature, as in the 2 pixel case a $90°$ shift of direction is possible within 2 steps of the detection. Within each cycle of the detection algorithm pixels that have been assigned to a segment are removed from

the input binary map to prevent double assignments. This procedure is repeated until no new starting cells are found.

After removing all linear segments, the remaining binary map contains only non-LKF pixels or LKFs forming closed contours with no starting points. The closed contours are opened by arbitrarily marking pairs of two neighbouring LKF-pixels (every $(i \cdot 100)$-th and $(i \cdot 100 + 1)$-th LKF-pixel for $i = 1, 2, 3, \ldots$) as starting points. Then the segment detection is repeated until no new starting points are found. The initialization step to open closed contours is then repeated until all LKF-pixels in

the binary map are assigned to a linear segment. All segments that were detected are shown in Figure 3 (a).

### 3.1.3 Reconnection

The reconnection instance is designed to connect multiple detected segments that belong to the same LKF. Two segments belonging to the same LKF should have a similar orientation and deformation magnitude and they should be in close proximity to each other. Thus, we compute the probability for all possible pairs of segments to be part of the same LKF based on their

distance, their orientation, and their deformation rates. The two segments of the pair with the highest probability are connected and the probabilities of pairs containing one of the two are updated. These steps are iterated until no new matches are found. This part of the algorithm represents a new feature compared to Linow and Dierking (2017).

The central element of the reconnection step is the probability matrix $P \in \mathrm{IR}^{N \times N}$ that stores the probabilities for all pairs of segments with N being the number of segments. The rows and columns correspond to single segments, where $P(m, n)$ gives

the probability of segment $m$ and $n$ being part of the same LKF. The probability is given by,

$$P(m,n) = \sqrt{\left( \left( \frac{\Delta D(m,n)}{D_0} \right)^2 + \left( \frac{\Delta O(m,n)}{O_0} \right)^2 + \left( \frac{\Delta \dot{\epsilon}(m,n)}{\dot{\epsilon}_0} \right)^2 \right)}, \tag{1}$$

with the elliptical distance $\Delta D$ between the two segments, the difference in orientation $\Delta O$, and the difference in total deformation rate $\Delta \dot{\epsilon}$. The difference in orientation $\Delta O$ is determined by the angle between the two segments, which are represented in this computation by a line connecting the start and the end point. The elliptical distance describes the distance between both

segments, but takes also into account the alignment of the segments. In doing so, we decompose the vector connecting both ends of the segments $v_{a \to b}$ into a orthonormal basis with one vector parallel to the first segment $a_{\parallel}$ and one vector perpendicular




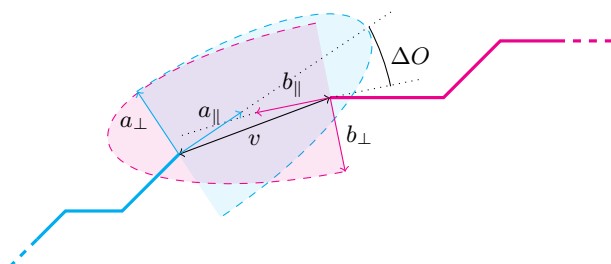

**Figure 2.** Sketch of two segments illustration the principle behind the elliptical distance. In the distance computation the component pointing in the direction perpendicular to the segment is weighted by a factor of $e$. Within the shaded area the elliptical distance is below a threshold $D_0$. If the endpoints of both segments lay within the area where both shaded half-ellipse overlap, they are considered for reconnection.

to it $a_\perp$ as shown in Fig. 2. In the same way, $v_{b \to a}$ is decomposed in a basis for the second segment $b_\parallel, b_\perp$:

$$v_{a \to b} = \left(a_\parallel, a_\perp\right) \cdot \alpha \quad \overset{!}{=} \quad v_{b \to a} = \left(b_\parallel, b_\perp\right) \cdot \beta. \tag{2}$$

In the computation of the length of the connecting vector $v$, the component perpendicular to the segment is weighted with an elliptical factor $e > 1$ and the distance computed by both bases is averaged to obtain a symmetrical elliptical distance, that is,

$v = v_{a \to b} = v_{b \to a}$:

$$\|v\|_e = \frac{1}{2}\left(\|v_{a \to b}\|_e + \|v_{b \to a}\|_e\right) = \frac{1}{2}\left(\sqrt{\alpha^T \cdot \begin{pmatrix} 1 & 0 \\ 0 & e \end{pmatrix} \cdot \alpha} + \sqrt{\beta^T \cdot \begin{pmatrix} 1 & 0 \\ 0 & e \end{pmatrix} \cdot \beta}\right) = \Delta D. \tag{3}$$

We consider only pairs of segments where the starting point of one segment lies in the direction of the other segment, that is, $\alpha(0) \geq 0$ or $\beta(0) \leq 0$. Thus points with same elliptical distance lie on a half ellipse centered at the endpoint of the segment as denoted in Fig. 2. The computed probability for a generic pair of segments $P(m, n) = P(n, m)$ is symmetric, because both

the elliptical distance, the orientation difference and the deformation rate difference are symmetric. Thus, we only compute $P(m, n)$ with $m < n$, where $P$ is simplified as an upper diagonal matrix.

The parameters $D_0$, $O_0$, and $\dot{\epsilon}_0$ not only normalize the individual components of Eq. (1) but also serve as an upper threshold for these components. If for one pair the elliptical distance $\Delta D$, the difference in orientation $\Delta O$, or the deformation rate $\Delta \dot{\epsilon}$ exceed the threshold $D_0$, $O_0$, and $\dot{\epsilon}_0$ it is not considered for the reconnection. The threshold values will be determined in

Section 3.1.4 and are given in Tab. 1.

After initializing the matrix $P$, the pair of segments with the highest probability $P(m, n)$ is connected and the connected segment $(m \leftrightsquigarrow n)$ replaces the old segment $m$. Thereby the number of segments is reduced by one and the $n$-th row and $n$-th column are removed from the probability matrix $P$. The elements of the probability matrix that correspond to the segments $m$ need to be updated and thus the $m$-th row and $m$-th column of $P$ are reevaluated based on the new connected segment

$(m \leftrightsquigarrow n)$. This process is iterated, where in each iteration the pair of segments with highest probability is connected, until no pair is left that satisfies the threshold values $D_0$, $O_0$, and $\dot{\epsilon}_0$.



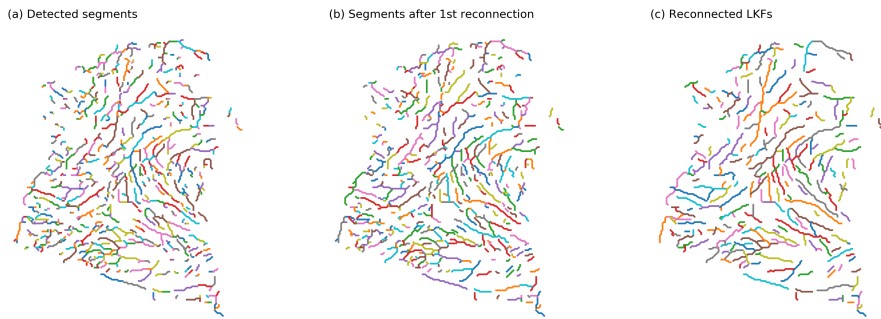

**Figure 3.** (a) Detected segments and the results of (b) the first reconnection step and (c) the second reconnection step for RGPS deformation data from January 1st, 2006. Each color only denotes a different segment or LKF. Due to a limited amount of colors, different segments or LKFs can have the same color. The output of the second reconnection step is the final output of the LKF detection algorithm.

In the final step of the LKF detection algorithm, features that fall below a minimum length $l_{min}$ are removed, because most small features are artifacts of the thinning algorithm and do not represent LKFs. With increasing minimum length, the number of detected LKFs decreases and the field of the detected LKFs shows a higher degree of abstraction. The minimum resolution of the detected LKFs, however, is determined by the minimum length used for the DoG filtering.

The presented reconnection procedure shows better results for longer segments, because the orientation and mean deformation is more sensitive for smaller segments. In theory, the best input would be segments containing all the points in the binary map that lie in between "junctions" of the lines, assuming that all those points belong to the same LKF. The segment detection instance, however, yields smaller segments due to the parallel detection that has been implemented to increase computational efficiency. Thus, we apply the reconnection algorithm twice: the first instance is meant to compensate for the tendency of the

segment detection algorithm to divide segments into smaller pieces although they actually belong to the same inter-junction segment. Thus, we use very a restrictive set of threshold values (maximum distance $D_0 = 1.5$, maximum difference in orientation $O_0 = 50°$, maximum difference in deformation rate $\dot{\epsilon}_0 = 0.75$, elliptical factor $e = 1$, and minimum length $l_{min} = 2$) to ensure that only segments are reconnected that are not separated by more than one pixel and no segments are removed because they are short (Fig. 3b). The second reconnection instance with a different set of parameters is then used to reconnect seg-

ments across junctions, and to generate the final LKFs shown in Fig.3 (c). The choice of set of parameters used in the second reconnection instance is discussed in Section 3.1.4.

### 3.1.4 Parameter optimization

There are a number of parameters in the detection algorithm. For some of them the range of possible choices can be narrowed down with information from field and satellite observations as well as theory of ice fracture, but none of them is strictly

constrained. Therefore, we attempted an optimization of the set of parameters, mainly of the reconnection step, given in Tab. 1.

    The main challenge of the optimization is the strong nonlinearity of the detection algorithm. The main source of non-linearity is the fact that whether a feature is detected or not can depend sensitively on small changes of a parameter. Another source



**Table 1.** List of parameters used in the LKF detection algorithm. For each parameter the lower and upper bound of the optimization is given along with the final choice of the parameter value.

| Parameter name | Symbol | Lower bound | Upper bound | Final choice |
|---|---|---|---|---|
| DoG filtering threshold | $d_{LKF}$ | 0 | 30 | $15^{\ddagger}$ |
| max. elliptical distance | $D_0$ | 1 | 7 | $4^{\dagger}$ |
| elliptical factor | $e$ | 1 | 5 | 2 |
| max. difference in orientation | $O_0$ | 25 | 65 | 35 |
| max. difference in deformation | $\dot{\epsilon}_0$ | 0.5 | 1.25 | 1.25 |
| min. length | $l_{min}$ | 3 | 7 | 3 |
| min. radius of DoG | $r_1$ | | | $1^{\star\dagger}$ |
| max. radius of DoG | $r_2$ | | | $5^{\star\dagger}$ |

[\*] parameters that have not been optimized but taken from Linow and Dierking (2017)

[†] parameters are related to length scales and need to be scaled with the spatial resolution of the input data

[‡] parameters are used to suppress noise in the input data and need to be adapted individually to input data

of non-linearity is the small amount of reference data. The strong non-linearity of the problem constrains both the definition of the cost function and the optimization method to make sure that the optimized solution is the global minimum of the cost function.

We constructed different cost functions, ranging from simple counts of falsely undetected features by the algorithm to a cumulative Modified Hausdorff Distance (MHD) between all detected and hand-picked features as a cost function, in an effort to smooth the strong non-linearity. In addition, we used different non-linear optimization routines including basin-hopping (Wales and Doye, 1997) and a nested brute-force implementation. No combination of cost function and optimization method lead to a satisfying result, because in all cases the cost function was very sensitive to smaller variations of the parameters. We concluded that finding a global minimum of the cost function is impossible. Therefore, we use a set of parameters estimated with a simple brute-force algorithm that minimizes the number of not-detected features for the range of parameters given in Tab. 1, where for each parameter five equally spaced values within its range are used. We do not regard this set of parameters as the global optimum, but rather as a useful working basis given the strong non-linearity of the problem and the limited amount of reference data. The performance of the detection algorithm with this set of parameters is evaluated in detail in Section 3.2.

### 3.2 Evaluation

In the evaluation we use all data for January 2006 (11 snapshots) from the hand-picked LKF data set, the LKFs detected by the algorithm presented in this study, and LKFs detected by the original version of this algorithm (Linow and Dierking, 2017). The reference data of the original algorithm is generated with the parameters used in the evaluation section of Linow and Dierking (2017). In this way, we evaluate the overall ability of the method to properly detect LKFs but also check whether the modifications and new additions improve the performance of the algorithm. We determine to which degree the algorithms





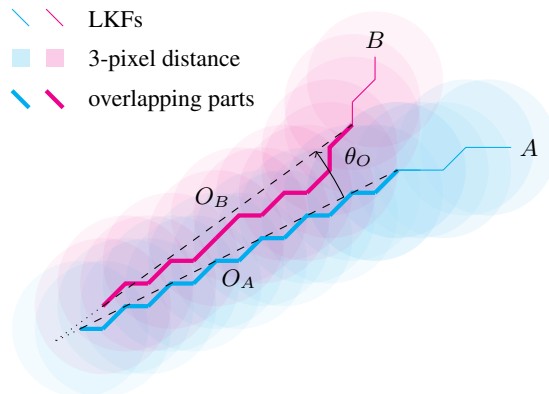

**Figure 4.** Illustration of overlap between two LKFs $A$ and $B$.

can detect the same features that were recognized by visual inspection and furthermore provide detailed information about the similarity and differences between automatically detected and hand-picked features in an element-wise comparison.

The principle idea behind this evaluation is that we compare the features pairwise: one LKF from the hand-picked data with the best-matching automated detected LKF. We find the best matching automatically detected feature for each hand-picked

feature by minimizing the Modified Hausdorff Distance (MHD) (Dubuisson and Jain, 1994) between the hand-picked features and all automatically detected features. Next, we categorize all pairs by the degree of overlap of the hand-picked feature with its closest matching detected feature. The overlap between two features is illustrated in Fig. 4. The number of pixels for which the distance to the closest pixel in the matching feature is smaller than $d_O = 3$ is defined as overlap, labeled as $O_A$ and $O_B$ in Fig. 4. To distinguish between the overlap of feature pairs that have similar shape but are displaced and pairs that overlap only

due to intersection of both features, we compute the angle between overlapping parts of the matching pairs $\theta_O$. If this angle is smaller than $\theta_O < 25°$, the overlap of a matching pair is defined by the minimum of overlapping pixels of both matching partners normalized by the maximum length of both matching partners:

$$O = \min(\text{len}(O_A), \text{len}(O_B)) / \max(\text{len}(A), \text{len}(B)), \quad \text{for} \quad \theta_O < 25°. \tag{4}$$

Given this definition of overlap, we distinguish between three different classes of pairs: (1) *fully matching pairs* that have an

overlap larger 60%, (2) *partly matching pairs* that have an overlap smaller then 60% but larger than 0%, and (3) *not matching pairs* with overlap equal to 0%.

The overall performance of the algorithm with respect to the overlap of the detected features with the reference data-set is given in Fig. 5 along with the number of pairs within each class. We find that the features detected with our new algorithm overlap significant more with the hand-picked reference data than the features detected with the original version of the algorithm

(Fig. 5). The original version of algorithm is run with the same parameters used in the evaluation section of Linow and Dierking (2017). Our modifications to the algorithm increase the number of fully matching LKFs by 66% (from 314 to 522), along with



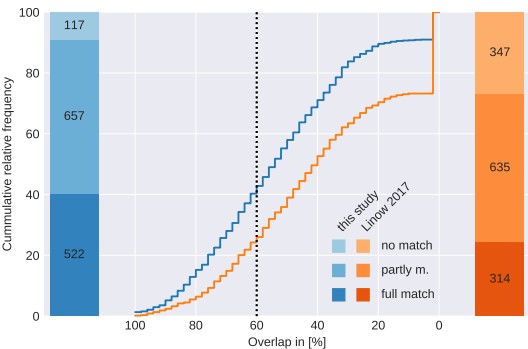

**Figure 5.** Evaluation of detection algorithm presented in this study and original algorithm (Linow and Dierking, 2017) against hand-picked LKFs. The cumulative frequency of occurrence of the overlap is given in the center plot. The amount of features is given in the bar plots for each class (fully matching, partly matching, and not matching).

a similar amount of partly matching LKFs (from 635 to 657), and a clear decrease of 66% for the not matching LKFs (from 347 to 117). This indicates a significant improvement of the original algorithm.

We analyze the similarity of features of all pairs within each class to test whether these improvements are made at the expense of the quality of the detected features. In doing so, we define three metrics to determine the similarity of the features: (1) the

mean endpoint distance, (2) the line length error, and (3) the MHD as metrics, where the first two were introduced by Linow and Dierking (2017). The MHD is a measure of the general agreement of two shapes. It takes into account changes in both orientation and length, but also a complete change in shape. In the sea ice context, the MHD is applied, for example, to evaluate the ice edge position in sea ice forecasts (Dukhovskoy et al., 2015) or to assess the predictability of LKFs (Mohammadi-Aragh et al., 2018). We here only focus on the class of full matching pairs.

For the mean endpoint distance, we determine the distance between both endpoints of the detected and the hand-picked feature for each pair and average them. For all full matching pairs with features detected by our new algorithm the endpoint distance tends to be smaller compared to features detected by the original version of the algorithm, which is indicated by the shift in the distribution towards smaller errors (Fig. 6a). The improved match because of the modifications to the original algorithm is also reflected in the smaller mean error (1.47 px as opposed to 1.96 px of the original algorithm). For 75% of the

features detected with our algorithm, the mean endpoint distance is smaller than 2 px whereas this is only the case for 60% of features detected by original version.

The line length error is determined by the difference in length of the two features in a pair normalized by length of the smaller feature of the pair. For both algorithms, the distributions are similar with a slightly smaller mean error of 18.54 % as opposed to 18.74 % for the original algorithm (Fig. 6b). For half of the pairs the line length error is also lower than 15% in

both cases.



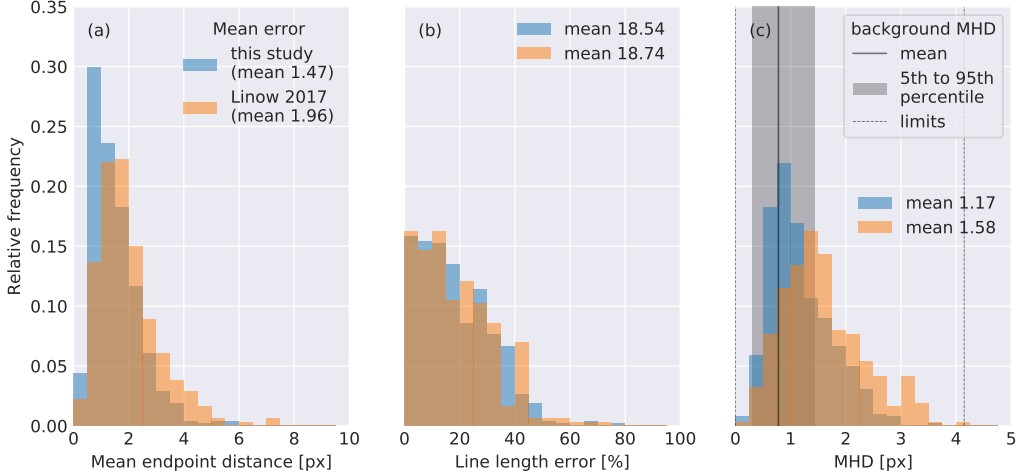

**Figure 6.** Statistics of all full matching pairs computed for the algorithm presented here and the original version (Linow and Dierking, 2017): (a) the mean endpoint distance, (b) the line length error, and (c) the Modified Hausdorff Distance (MHD). The background MHD refers to to MHD calculated for the hand-picked features and the morphological thinned binary field (Fig. 1f).

We find that our modifications to the original algorithm also reduce the average MHD from $1.58$ px to $1.17$ px (Fig. 6c). $73\%$ of these pairs lie within the 5th to 95th percentile of the background MHD defined as the MHD of the reference data and the morphological thinned binary LKF field (Fig. 1f). Since all LKFs consist of sets of pixels from this binary field, the background MHD is an upper limit of how accurate a reconnection algorithm can get using this binary field as an input value.

In conclusion our new version of the algorithm improves the original algorithm in that it detects more features and also increases their agreement with the hand-picked reference data.

### 3.3    Discussion

Our adapted detection algorithm greatly improves the original version. The total number of hand-picked features that is reproduced by the algorithm increased by $66\%$. In addition, also the quality of the detected features with respect to their mean

endpoint distance, the error in line length and the MHD is improved. We attribute these improvements to two changes that stand out besides smaller adjustments in the code: (1) the introduction of a probability based reconnection and (2) the optimization of the DoG threshold $d_{LKF}$, which showed the highest sensitivity in the optimization. We use this threshold to filter LKFs as regions that have high deformation rates compared to the local environment. The deformation rates in RGPS are known to be prone to grid scale noise (Bouillon and Rampal, 2015), which can lead to a false classification of a pixel as LKF. Increasing

the threshold sightly suppresses this noise, albeit at the expense of loosing features with smaller deformation rate differences. Thus the threshold needs to be optimized to balance both effects; we found $d_{LKF} = 15$ to be the best parameter choice for the RGPS data set.





In our algorithm we reconnect segments to LKFs based on a probability computed from the characteristics of the segments. Thereby those segments are reconnect that fit "the best" and in contrast to Linow and Dierking (2017) the reconnection does not depend on the order in which the reconnection algorithm runs over the list of segments. In doing so, we improve the quality of the detected features and obtain a unique and consistent solution. Both uniqueness and consistency are necessary ingredients

for the ensuing application of a tracking algorithm.

In this context, we found the elliptical distance $\Delta D$ and the orientation $\Delta O$ to be the important contributers to the probability function. The optimized threshold $\Delta\dot{\epsilon}_0 = 1.25$ for differences in deformation rates is very high (as it is applied to the difference of the common logarithm of the deformation rates, a difference between deformation rates of a factor $10^{1.25} = 17.78$ is possible), so that differences in deformation rates are normalized by a large value and do not contribute much to the probability

function. Omitting the deformation related part in Eq.1 (equivalent to setting $\Delta\dot{\epsilon}_0 = \infty$) does not change the results of the evaluation of the optimized solution very much (not shown here). The small influence of the threshold for deformation rates $\Delta\dot{\epsilon}_0$ on the performance of the algorithm may also be caused by the noise in the RPGS data along LKFs (Bouillon and Rampal, 2015). Especially smaller segments are affected by the noisy RGPS data so that segments that belong to the same LKF may have different deformation rates.

## 4   LKF tracking

The dynamic nature of the ice pack with spontaneous fracture, fast propagation of failure lines, and discontinuous drift fields makes tracking of deformation features in the ice a challenge. Because most of these processes occur on time scales from seconds to days, the temporal resolution of the RGPS data set of 3 days makes feature tracking even more challenging. In this section, we present an algorithm that automatically tracks features and we compare the tracked features to hand-tracked

features.

### 4.1   Method description

The tracking of deformation features, in fact any feature, between two time records is always a two-step problem: first, the deformation features are detected for both records separately and second, the features of both time records are connected in time by identifying features of the first record with those of the second record.

Between two RGPS time records (3 days) a deformation feature will be advected and can undergo the following changes: (1) it can become inactive, (2) it can grow, or (3) shrink, or it can undergo a combination of growing and shrinking. Thus, on top of two time records, tracking requires drift information between these records. From the same drift fields that were used to derive the deformation data we estimate an first-guess position of each feature from the first record in the second record that neglects all effects but advection. We compute the drift first-guess position in pixel space (each feature in the first record is described

by integer pixel indices) by normalizing the drift speed with the grid resolution. Thus, the computed first-guess positions are given in floating point indices of the input field of the detection algorithm.



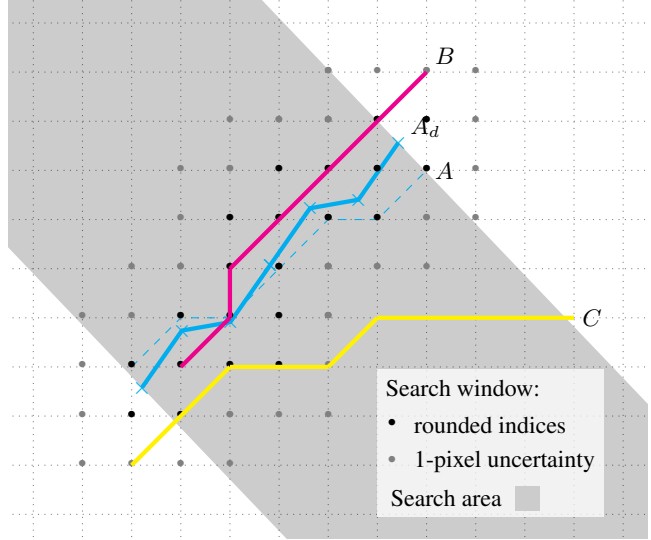

**Figure 7.** Principle of the tracking algorithm showing the search area and search window. $A$ is the original feature (dashed blue) and $A_d$ (blue) the first-guess position considering only drift. $B$ and $C$ are two features in the second record, where $B$ is marked as a successfully tracked feature.

For the following description, a tracked feature is a feature from record two with an associated feature in record one; a matching pair is a pair of associated features from records one and two; and all matching pairs are called the tracks.

A tracked feature in the second record is required to overlap at least in part with the first-guess position after growing and shrinking in between the time records is taken into account. We define a search window around the first-guess position of the feature to test for an overlap of the features with the first-guess position. The search window consists primarily of pixels for

which the floating point indices of the first-guess position are rounded up and down by the python functions `ceil` and `floor`. To take into account the position uncertainty caused by the morphological thinning algorithm, we also add all neighboring pixels of the pixels with rounded indices using the mean background MHD of the morphological thinned field (Fig. 6c) of 0.78 px as an estimate for this uncertainty. All features in the second record that include a minimum number of pixels $o_{\min} = 4$

of the search window are marked as potentially tracked features. If we consider a feature that changes shape only due to advection without any growing or shrinking, the tracked feature from the subsequent time record should lie completely within the search window.

During the course of three days, however, many features grow or an opening closes at one or both ends of the detected feature. Also, in rare cases, only parts of a feature close and a new branch is formed within the position of the old feature.

This will be referred to as branching. Our algorithm is designed to take into account only growing and shrinking, because to our experience there are rather few branching LKFs (10 %) and because branching is very complex to track. Thus, a feature that is considered as tracked feature is allowed to grow at both ends compared to the first record or to shrink to only a part of the original feature. To translate this into an algorithm, we define a search area that is the area enclosed by two lines through



the endpoints of the first-guess position. These lines are perpendicular to the orientation of the first-guess position (see grey shaded area Fig. 7). For a tracked feature, all points of this feature that lie within the search area need also to lie within the search window. Here, we implement a threshold value $p_{w/a}$ that defines the fraction of points within the search window and points within the search area for the feature to be considered as a tracked feature. Features, for which more pixels lie within

the search area but not in the search window, have likely undergone branching or just intersect with the first-guess position but have a different orientation.

The last step of the tracking algorithm filters small features inside the search window that intersect with the first-guess position. Due to their short length all of their points in the search area also lie in the search window. To exclude those we compute the overlap as defined in Sec. 3.2 between the first-guess position and the potentially tracked feature. We use a

maximum distance of pixels of $d_O = 1.5$ and an angle threshold of $\theta_O = 25°$ for the computation of the overlap. All potentially tracked features with a non-zero overlap are marked as tracked features.

For all features of the first record this procedure is repeated iteratively: (1) advecting the feature using the drift information to obtain the first-guess position (2) check for features that share $o_{min}$ pixels with the search window, (3) compute the fraction of pixels in the search window and in the search area $p_{w/a}$, and (4) test for non-zero overlap. The output of the tracking algorithm

is a list of matching pairs of always one feature from the first record and a tracked feature from the second time record.

### 4.2   Parameter optimization

In the tracking algorithm the four parameters $o_{min}$, $p_{w/a}$, $d_O$, and $\theta_O$ are not very well constrained, so we attempt to optimize them within plausible bounds. As we want to optimize the tracking algorithm independently of the detection algorithm, we use the hand-picked features as input for the tracking algorithm and compare the output to the hand-tracked features. We perform

the same very basic optimization as in Sec. 3.1.4 facing similar problems with limited amount of reference data and a non-linear cost function. We choose equally spaced values in the range given in Tab. 2 for all four parameters and determine the number of correctly tracked features, the missed tracks, and false positives. We find that decreasing $o_{min}$ and increasing $p_{w/a}$, $d_O$, and $\theta_O$ leads to an increase in correctly tracked features along with a large increase of false positive. To balance both effects, we define the cost function as the number of correctly tracked features subtracted by the number of missed tracks and the number

of false positives. The final parameter set that maximizes this difference is given in Tab. 2.

### 4.3   Evaluation

To separate the two steps of feature tracking, and to enable an independent evaluation of the tracking algorithm, we apply the tracking algorithm on two different LKF data-sets: the hand-picked features and the features extracted by the detection algorithm for the same time span. Then, we compare both results to the hand-tracked features described in Sec. 2.3.

We evaluate the tracking algorithm independently by applying it to the hand-picked features and test whether the algorithm reproduces the hand-tracked features of the next time record, hereafter referred to as "only tracking". The algorithms picks 336 (85.7%) of the overall 392 hand-picked tracked features correctly and only misses 56 (14.3%). Besides the missed tracks, the algorithm detects 89 false positives. We use the information describing the type in change of shape (no change, growing,





**Table 2.** List of parameters used in the LKF tracking algorithm. For each parameter the lower and upper bound of the optimization is given along with the final choice of the parameter value.

| Parameter name | Symbol | Lower bound | Upper bound | Final choice |
|---|---|---|---|---|
| min. overlap in search window | $o_{\min}$ | 2 | 6 | 4 |
| fraction of pixels in search window and in search area | $p_{w/a}$ | 0.5 | 1 | 0.75 |
| max. distance of pixels for overlap | $d_O$ | 0.75 | 2.25 | 1.5 |
| max. angle for overlap | $\theta_O$ | 15 | 35 | 25 |

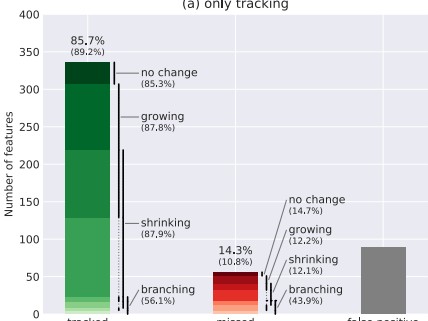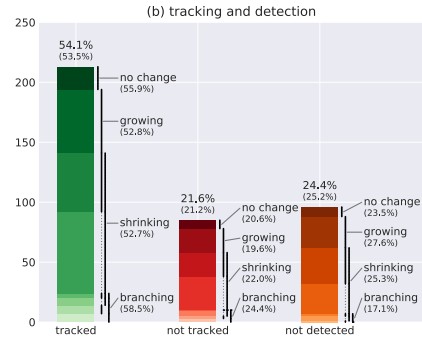

**Figure 8.** Evaluation of (a) only tracking algorithm on the hand-picked features and (b) combination of detection and tracking algorithm. The color segments of each bar shows the different combinations of changes in shape (no change, growing, shrinking, and branching) that is labeled by the black lines next to the bar. The missed tracks in (b) are separated into not detected or not tracked features. Above the bars the percentage compared to all hand-picked tracks is given for all types and for all types except branching, which is given in brackets.

shrinking, and branching) provided for the hand-picked tracks to test the performance of the algorithm for those different types of change (Fig. 8a). The performance of the algorithm ranges from 85% to 88% for the different types with the branching type being an exception, for which only 56% of the features are tracked correctly. This is not surprising as the algorithm is not designed to track this type of change.

5     In the evaluation of both the detection and tracking algorithm, we distinguish between missed tracks that where not tracked by the tracking algorithm and missed tracks, for which the detection algorithm was not able to detect the corresponding features in both time records. First, we test whether the detection algorithm picks both features of a hand-picked matching pair. In doing so, we separate both features into the parts that both features share and a non-overlapping part, to account for varying shapes in the non-overlapping part of detected and hand-picked feature. Then, we check whether detected features correspond to the

10    hand-picked features using the overlap as in Sec. 3.2. If for one of the two hand-picked features no corresponding detected feature is found, this hand-picked tracked feature is marked as a missed tracked feature caused by the detection algorithm. Otherwise, we test whether the tracking algorithm tracks the detected features appropriately. In total, 54.1 % of the hand-picked





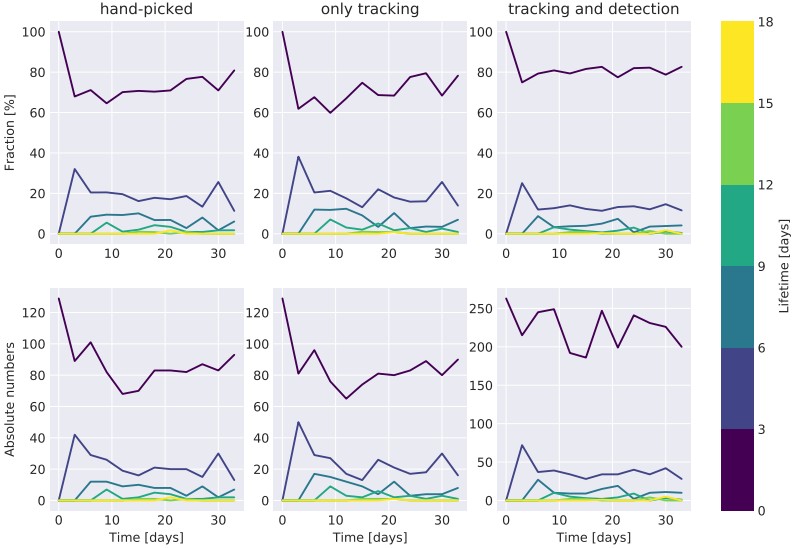

**Figure 9.** Distribution of lifetime of all features in the first 12 RGPS snapshots in 2006 in the hand-picked reference data (left column), the automatically tracked features of the hand-picked features (middle column), and the automatically tracked features of the automatically detected features (right column). The lower row shows the absolute number of features for a certain lifetime class, whereas the upper row is normalized by the total number of features for this time record.

matching pairs are detected and tracked correctly, whereas 21.6 % are not captured by the tracking algorithm, and for 24.4 % of the tracked features the corresponding features are not detected. Interestingly, these fractions do not change significantly if sub-sampled to individual types of change. Only for the branching type of tracks, the rate of tracked features missed by the tracking algorithm exceeds the one for the detection algorithm, which is in line with the low amount of matching pairs captured by the tracking algorithm for this type of change found in the evaluation of the tracking algorithm alone.

Since not all hand-picked tracked features are tracked automatically by the presented algorithm, we need to make sure that this does not change the temporal characteristics of the automatically generated tracked features. In doing so, we compare the distributions of lifetimes (Fig. 9) and growth rates (Fig. 10) of the hand-picked tracked features (hand-picked), the automatically tracked features of the hand-picked features (only tracking), and the automatically tracked features of the automatically detected

features (tracking and detection). At the beginning of the evaluation period, all features are initialized with a lifetime in the class of 0 to 3 days, where the range of the class is given by the temporal resolution of the input data. The following time steps, all features that are marked as tracked features are assigned to the next lifetime class compared to the class of their tracking partner in the previous time record. All other features are initialized again in the lowest category 0 to 3 days. For all three different data sets, more than 99% of features have a lifetime smaller than 12 days, which can be thought of as the average

spin-up time needed by the tracking algorithm. In the following, we consider only the period after those 12 days.





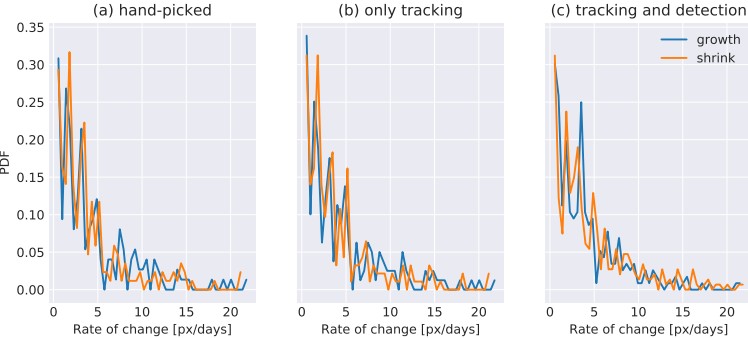

**Figure 10.** Probability distribution function for growth rates in (a) the hand-picked reference data, (b) the automatically tracks of the hand-picked features, and (c) the automatically tracks of the automatically detected features separated in growth for positive growth rates and shrink for negative values.

The distribution of lifetimes for the hand-picked tracks and automatically generated tracks of hand-picked features are very similar: (1) the number of features in the lowest lifetime category increases in absolute and relative numbers to the end of the evaluation period in a equal manner (70 % to 81 % (hand-picked) and 67 % to 78 % (only tracking), (2) 17 % (hand-picked) and 18 % (only tracking) of the features have a lifetime of 3 to 6 days, and (3) for both the remaining 9 % of the features have

a lifetime over 6 days. In a feature-by-feature comparison for both data sets, only 10 % of the features vary in their lifetime due to uncertainties in the tracking algorithm. The root mean square error of the lifetime is estimated to be 1.55 days. For the automatically detected and tracked features the average lifetime reduces to 3.9 days compared to 4.2 days for the hand-picked features, which is driven by an increase in the amount of features in the lifetime class 0 to 3 days to 81 %. The fractions of the remaining classes are reduced accordingly, but do not change significantly relative to each other.

Besides the lifetime as the main temporal characteristic, we compute the growth rates of all tracked features, to check whether those change in a similar manner in shape. The growth rate is defined as the difference of the number of pixels of the feature in the second record compared to the feature of the previous record. In our analysis, we divide the growth rate into two regimes depending on their sign: growth of the feature for positive values and shrinking of the feature for negative values. The distributions of the growth rates for the "hand-picked", "only tracking", and "tracking and detection" data sets

all have an exponential distribution (Fig.10) with half of the features changing by less than 3 pixel per day. The high order of similarity of the distributions indicates that the usage of the detection and tracking algorithm does not distort the characteristics of tracked features, even though the total number of features detected and tracked increases by a factor of 2.3 for the detection and tracking data-set.

## 4.4  Discussion

The large time difference of 3 days between records of the RGPS data set significantly complicates the tracking of deformation features because they change shape and positions on shorter time-scales. With this background, the overall percentage of 85.7 %



of hand-picked tracked features that were correctly identified by the algorithm is more than satisfying. The missed tracks and the false positives of the algorithm lead to an lifetime RMSE of 1.55 days, which is smaller than the uncertainty given by the 3 day temporal resolution of the input deformation data. The low percentage of correctly identified branching features is also acceptable because they only make up 10 % of all hand-picked features. We hypothesize that relaxing the constraints in the

algorithm to also track branching features will most likely lead to a strong increase in false positives.

Also the combination of tracking and detection algorithm reproduces more than half of the hand-picked tracks. This exceeds the 40 % of features that where fully detected by the detection algorithm and might hint at a better performance of the algorithm for long-lived features. The hand-picking of features and tracks by only one individual also leads to a bias in the reference data. To accurately separate the uncertainty caused by the subjectiveness of the reference data from the uncertainty of the

algorithm, more individuals would need to repeat the hand-picking procedure, which would exceed the scope of this manuscript. Especially, small LKFs and LKFs in regions of low deformation are harder to catch by eye (Linow and Dierking, 2017), which explains that the automatic detection picks 2.3-times more features than in the reference and the number of tracked features increases by 65 %. We assume that this bias towards small and therefore most probably short-lived features is responsible for the slightly higher percentage of features in the class with the lowest lifetime (0 to 3 days). Besides this small increase, the

distributions of lifetimes for the automatically detected and tracked features are very similar, which suggests that there is no significant cumulative bias caused by the application of both algorithms. This is backed by the similar growth rates observed for both data sets.

## 5  LKF data-set

### 5.1  Generation of LKF data-set

In this section, we introduce a data-set of LKFs generated by applying the detection and tracking algorithm to all available RGPS data. The RGPS data covers the time from November 1996 to April 2008. Here, we use the available winter data from late freezing season (November/December) to the start of the melt onset (April/May). The Lagrangian drift information that is also provided in the RPGS data set is interpolated to the regular grid used for the RGPS deformation data to be used as input in the LKF tracking.

First, we apply the detection algorithm with the optimized parameters given in Tab. 1 to all deformation data. The output of the detection algorithm is a list of LKFs for each time record that includes one array for each LKF. The array stores the position (as index in the RGPS grid and in lat/lon-coordinates) and deformation (divergence and shear rate) information of all points of the LKF. Next, we feed the interpolated drift information and the detected LKFs of each year to the tracking algorithm and determine the linkages between LKFs for successive time records. The tracking algorithm with the optimized parameters given

in Tab. 2 provides a list of tracked pairs. Each pair contains the indices of the LKFs in record one and two.

Overall 164 698 LKFs were detected and 35 855 tracked features were found. The yearly detection numbers range from 11 002 LKFs for the winter 2006/07, the year of a sea-ice minimum, to 16 774 LKFs in winter 2001/02. If the number of



detected features is normalized by the number of observations of sea-ice deformation, we find the maximum for winter 1996/97 and the minimum in 2002/03. The number of tracks vary from 2 012 tracks in winter 2003/04 to 4 017 tracks in winter 2001/01.

## 5.2 Applications and Discussion

In this section, we present a few illustrative statistics of the LKF data-set. In doing so, we intend to demonstrate the usefulness of the data-set but also to check it for consistency with other studies on leads and sea-ice deformation. The statistics (Fig. 11) range from spatial properties such as LKF length, LKF density and intersection angles, to temporal properties such as LKF lifetimes. In case of the intersection angle, we give an example for a model-observation comparison with the presented algorithms by comparing the RGPS LKF data-set to LKFs that were detected and tracked in a 2-km model simulation with a numerical sea-ice ocean model. In addition, we link the number of deformation features and its corresponding deformation rates to atmospheric drivers, in particular to Arctic cyclones.

The density of LKFs is computed for the all years of the LKF data set as the incidences when a pixel is crossed by a LKF normalized by the overall number of RGPS observations for this pixel. In Fig. 11 (a) only pixels that have more than 500 RGPS observations are shown. We observe a fairly homogeneous LKF density throughout the entire Amerasian basin, with a slight increase in the Beaufort Sea. The fast ice regions in the East Siberian Sea have the lowest densities with the fast ice edge showing up as a sudden increase in LKF density. The highest LKF densities are found around the New Siberian Islands, Wrangel Island, and at the coastlines along the Beaufort Sea. This agrees very well with studies on lead densities derived from MODIS thermal-infrared imagery (Willmes and Heinemann, 2016) and CyroSat-2 data (Wernecke and Kaleschke, 2015) that show high densities in the Beaufort sea and highest values close to the coastline. A direct comparison of density values is not possible as those studies are limited to leads that are identified as an opening in the ice cover, whereas our algorithm picks regions of high deformation rates that can also include pressure ridges.

The distributions of LKF length of all single years are very similar and range from $\sim 50\,\text{km}$ to $1000\,\text{km}$ (Fig. 11c). The PDF is described accurately by a power-law for LKF length between 60 and $600\,\text{km}$. Due to the varying coverage and especially smaller gaps in the RGPS data, long features might have been divided into multiple smaller segments, which explains why the power-law does not hold for large LKFs. The smallest feature detected by the algorithm is $3\,\text{px} = 47.5\,\text{km}$, leading to the deviations from the power law at the smallest scales. The power-law scaling of LKF length fits into the picture of the distributions of lead width (Wernecke and Kaleschke, 2015) and deformation rates (Weiss, 2013) that also follow a power-law scaling.

The intersection angle between two deformation features formed at a similar time is related to the rheology describing the deformation of sea-ice. From satellite imagery in the visible range for 14 days in 1991, the intersection angle was found to range between $20°$ and $40°$ (Walter and Overland, 1993), which can be linked to the angle of internal friction using a Mohr criterion for failure (Erlingsson, 1988). The distribution of intersection angles of two LKFs that formed in the same time record is given in Fig. 11 (d), where only LKFs larger than $10\,\text{px} = 125\,\text{km}$ are taken into account to reduce the effect of a preferred direction ($45°$ and $90°$) originating from the rectangular grid. We perform this analysis on the RPGS LKF data-set and LKFs that were detected in a 2-km Arctic numerical model simulation, whose details are given in A. For the RGPS data, we find that





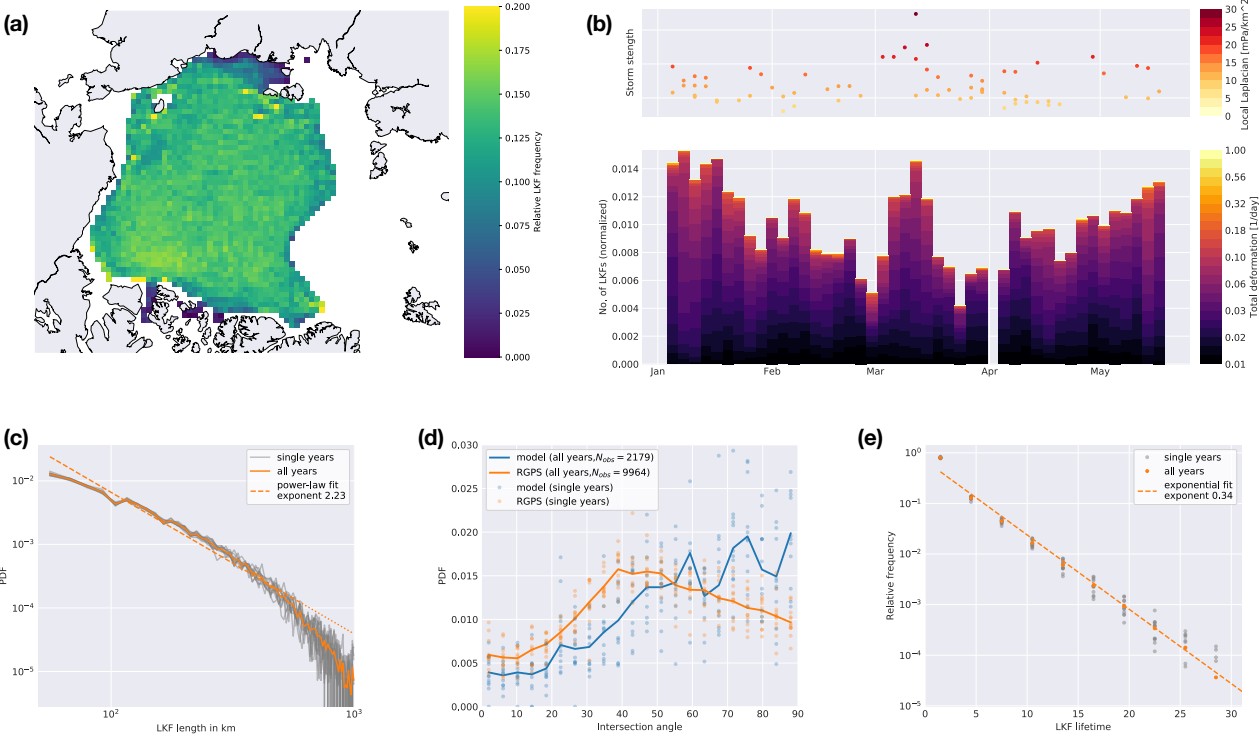

**Figure 11.** Statistics on LKF data-set: (a) LKF density computed for the all years of the data-set. (b) Time series of the number of LKFs separated by their deformation rates for winter 2002/03. For each time-step the number of cyclones (Serreze, 2009) over the Arctic ocean is shown in the upper panel along with the local Laplacian as a proxy for their strength. (c) PDF of LKF length along with power-law fit for LKFs smaller than 600 km. (d) Intersection angle of pairs of features that are new formed (lifetime between 0 and 3 days) and have size of at least 10 pixels for the RGPS and a 2-km Arctic numerical model simulation. (e) Distribution of LKF lifetimes for all years along with a fit to an exponential distribution.

the distribution peaks at an angle of $40° - 50°$. Angles larger $50°$ occur more often than angles below $40°$, and angles between $0° - 20°$ have the lowest occurrence. The broad distribution indicates that there is not only one specific fracturing angle, but that heterogeneities in the ice cover and temperature variations (Schulson, 2004) as well as the dilatancy effect (Tremblay and Mysak, 1997) may influence the deformation on an Arctic-wide scale. The LKFs in the model simulation intersect at larger

5  angles with local maxima in the range $60°$ to $90°$. The difference in the sample size might cause these strong variations for larger angles, as we find five times less features in the model simulation than for RGPS within the same period of time. In general, the model underestimates the probability of intersection angles smaller $55°$ and overestimates those of angles larger than $55°$. We attribute these differences to the usage of the elliptical yield curve with normal flow rule in the simulation, because





this yield curve does not have a "preferred" direction of fracture in contrast to yield curves with a Mohr-Coulomb criterion. The intersection angle may be improved by an appropriate choice of model parameters (Ringeisen et al., 2018).

The distribution of lifetimes determined by the tracking algorithm shows a clear exponential distribution with an exponent of 0.34 (Fig. 11e). Kwok (2001) described some LKF systems that were persistent in the Arctic over a period of a month for the winter of 1996/97. We also find lifetimes as high as this, but show that the majority of LKF are active in much shorter time intervals. We assume that the rapid changes in external forcing (mainly wind stress) are the reason for the high amount of short-lived LKFs.

Last, we study the link between the detected features and the wind forcing being the main driver of ice fracture. To do so, we combine a data-set of cyclones in the Northern hemisphere (Serreze, 2009) with the distribution of LKFs in different deformation rate classes for, as an example, the winter of 2002/03 (Fig. 11). In the freezing season, we find more deformation features (with general higher deformation rates) caused by the thinner and therefore weaker ice during this period. With thicker ice the number of deformation features decreases followed by an increase from April onwards which we attribute to the onset of melting and the resulting weakening of the ice. This overall seasonal cycle is interrupted by a set of four strong cyclones that pass the Arctic oceans in March and lead to a sudden increase of the number deformation features. This confirms that weather systems with high wind speeds are a main driver of sea-ice deformation in the Arctic ocean.

## 6 Conclusions

The new detection algorithm presented in this study follows the structure of the original algorithm (Linow and Dierking, 2017) with classifying LKF pixels in the input deformation rates and then detecting single deformation features. In doing so, an additional degree of abstraction is added compared to studies using only skeletons of leads (Banfield, 1992; Van Dyne and Tsatsoulis, 1993; Van Dyne et al., 1998). This enables not only the extraction of feature-based information such as intersection angles and LKF length but also the tracking of the features. In addition, avoiding classified, that is, preprocessed and charted, SAR imagery (Banfield, 1992; Van Dyne and Tsatsoulis, 1993; Van Dyne et al., 1998) provides the opportunity to apply the algorithm to both model output and satellite observations. For instance, Koldunov et al. (2018) applies the algorithm directly to sea ice thickness as an input field to study the impact of solver convergence on the cumulative effect of deformation features. Still, the algorithm can also be applied to classified imagery, if the first filtering steps are skipped and the classified imagery is used as the binary LKF map (like Fig 1e).

The evaluation of the detected features shows that introducing a probability based reconnection instance improves both the number of correctly detected features and their quality. Here, the input of distance and differences in orientation are the most important contributions if we consider the high threshold for difference in deformation resulting from the parameter optimization. We only performed a brute-force optimization of the parameters of the detection algorithm for a small parameter space limited by the strong non-linearity of the detection itself and the small amount of reference data. For a thorough optimization a larger reference data-set is required.





The design of a new tracking algorithm is outlined. The tracking algorithm handles the dynamic nature of sea ice as well as the low temporal resolution of satellite drift data. The algorithm takes advection as well as growth and shrinking of deformation features appropriately into account (86 % of the hand-picked tracks are found correctly). The algorithm recognizes the opening of secondary leads (branching) at a lower rate (56 %), but one needs to bear in mind the higher uncertainty of those

features in the hand-picked data set and their generally smaller number. The performance of the combination of detection and tracking algorithm is also satisfactory and does not bias the statistics of the features. Roughly 20-30 % of the detected features are tracked. Consequently, the remaining 70–80% of the features persist for less than 3 days. Sea-ice deformation at higher sampling rates, for example, derived from ship radar with a sampling rate of up to 10 minutes (Oikkonen et al., 2017), would be necessary to study LKF lifetimes at these shorter time scales.

We split the task of finding a deformation feature and following it with time in a spatio-temporal deformation data-set into two subroutines: (1) detection features in the deformation field of one time step and (2) finding the temporal connection between individual detected features of subsequent time steps. In doing so, both subroutines are independent of each other, although we speculate that information of the temporal evolution of sea ice deformation could in turn improve also the detection of features. For this task, machine learning techniques, which have recently attracted attention in the climate science context (see

for instance Ashkezari et al., 2006, for oceanic eddy detection), are a promising tool to explore.

The LKF data-set generated by automated LKF detection and tracking from the RGPS sea-ice deformation data includes $\sim 165\,000$ LKFs from 11 winters. These are significantly more deformation features than can be found in previous, hand-picked lead data sets (e.g. Miles and Barry, 1998). Due to the use of drift observation derived from SAR-imagery, the data-set is also not limited to clear-sky conditions. This object-based data set enables statistics of both the overall LKF field, like LKF density,

and of single LKFs, like length, intersection, curvature, etc. In addition, all of these statistics can be combined, linked and used as filter criteria. Along with the age estimated by the tracking algorithm, the data-set makes a comprehensive and quantitative description of deformation features in Arctic ocean possible and complements qualitative studies (Kwok, 2001).

The algorithms are designed in a flexible way so that they can be applied to any sea-ice drift and deformation data, or classified imagery. For example, the current RGPS LKF data-set could easily be extended until today with operational drift data

derived from Envisat and Sentinel-1 (Pedersen et al., 2015). Also, resolved leads in high-resolution Arctic model simulations can be analyzed to compare LKF properties to the LKF data set. We have shown a first example of comparing intersection angles of LKFs. Comparing the characteristics of deformation features directly makes a thorough evaluation of lead-resolving sea-ice models possible instead of focusing on only one property such as lead density (Wang et al., 2016) and also facilitates the complicated interpretation of scaling analysis of sea-ice deformation that has been used for this purpose so far (Rampal

et al., 2016; Hutter et al., 2018).

*Code and data availability.* The LKF data set derived from RGPS data will be available on pangea and the code of the LKF detection and tracking algorithm on github once the manuscript is accepted for publication.





## Appendix A: Details on the Arctic simulation

The Arctic simulation with a refined horizontal grid spacing $2\,\mathrm{km}$ using the MITgcm is based on a set of regional Arctic configuration (Nguyen et al., 2012). The number of vertical layers is reduced to 16 with the first five layers covering the uppermost $120\,\mathrm{m}$ to reduce computational cost as we are only interested in sea-ice processes. The Refined Topography data

set 2 (RTopo-2) (Schaffer and Timmermann, 2016) is used as bathymetry for entire model domain. The lateral boundary conditions are taken from globally optimized ECCO-2 simulations (Menemenlis et al., 2008). We use the 3-hourly Japanese 55-year Reanalysis (JRA-55) (KOBAYASHI et al., 2015) with a spatial resolution of $0.5625°$ for surface boundary conditions. The ocean temperature and salinity are initialized on January 1st, 1992 from the World Ocean Atlas 2005 (Locarnini et al., 2006; Antonov et al., 2006). The initial conditions for sea-ice are taken from the Polar Science Center (Zhang et al., 2003).

Ocean and sea ice parameterizations and parameters are from Nguyen et al. (2011) with the ice strength $P^\star = 2.264 \cdot 10^4\,\mathrm{Nm^{-2}}$. The momentum equations are solved by an iterative method and Line Successive Relaxation (LSR) of the linearized equations following Zhang and Hibler (1997). In each time step ($120\,\mathrm{s}$), 10 non-linear steps are made and the linear problem is iterated until an accuracy of $10^{-5}$ is reached, or 500 iterations are performed.

*Author contributions.* NH developed and implemented all modifications in the LKF detection algorithm. NH developed and implemented

the LKF tracking algorithm. NH performed the parameter optimization and evaluation for both algorithm. NH derived and analyzed the LKF data set. ML contributed to the analysis of the data set. LZ rewrote the original version of the algorithm in python as a basis for further developments. NH prepared the manuscript with contributions from all co-authors.

*Competing interests.* The authors declare that they have no conflict of interest.

*Acknowledgements.* We acknowledge Stefanie Linow and Wolfgang Dierking for help with the implementation of the detection algorithm,

the inspiring discussion on the development of the tracking algorithm and for their comments on the manuscript. We thank Khalid A. Maghawry for his support establishing the hand-picked tracking evaluation data-set.



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
