# Peer review of "Leads and ridges in Arctic sea ice from RGPS data and a new tracking algorithm"

_The Cryosphere, 2018_

## Referee Comment (RC1) · Anonymous Referee #1 · 22 Oct 2018

The authors have taken the sea-ice deformation data from the Radarsat Geophysical Processor System (RGPS) and used it to create a data set of Linear Kinematic Features (LKFs), i.e. leads and ridges that extend across the Arctic pack ice. Spatial coverage includes the western Arctic Ocean with 12.5 km grid size; temporal coverage is 1996-2008 during the winter season (Nov-Apr) with 3-day resolution. The LKFs are tracked over time using a new algorithm. Summary statistics of the LKF data set are presented.

The new data set could prove to be useful for calibration and validation of high-resolution sea-ice models, and as a climatology of LKFs for comparison with a future Arctic sea-ice cover. The paper is mostly a description of algorithms (pages 4-19), but its value lies in the end product – the LKF data set. In my opinion, the paper should be

published after the following comments are addressed.

=====

General Comments

Leads are not distinguished from pressure ridges in the data set – they are both labelled as LKFs. Wouldn't it be useful, once an LKF has been identified, to label it as "lead" or "pressure ridge"? Wouldn't it be possible to do this based on the values of divergence within the LKF, with positive divergence indicating a lead and negative divergence indicating a pressure ridge? Leads and pressure ridges are radically different types of features. Leads are the sites of new ice growth, contributing to the thin end of the ice thickness distribution. Pressure ridges affect air and water drag, and contribute to the thick end of the ice thickness distribution. Considering that 165,000 LKFs have been identified, wouldn't it be useful to know what percent of them are leads and what percent are pressure ridges? In my opinion, the authors should either go back and label each LKF as a lead or pressure ridge, or explain why it's impossible or too difficult to do so, or why it doesn't matter.

=====

Specific Comments (in page order)

Page 3 Section 2.1

(i) The RGPS data set needs a reference. Where does it come from? A NASA web site? The Alaska Satellite Facility?

(ii) It would be courteous to give credit for the creation of the RGPS, such as:

Kwok R. (1998) The RADARSAT Geophysical Processor System. In: Analysis of SAR Data of the Polar Oceans. Springer, Berlin, Heidelberg. https://doi.org/10.1007/978-3-642-60282-5_11

Page 3 Section 2.2

Again the source of the RGPS Lagrangian drift data should be cited.

Page 3 line 24

Please clarify that "px" = pixel and that 1 pixel = 1 grid cell = 12.5 x 12.5 km.

Page 4 line 14

After the equation, say that eps_I is divergence and eps_II is shear.

Page 4 lines 15-16

The LKFs are not "marked" by regions of high deformation, they are DEFINED by regions of high deformation. Consider writing: "LKFs are defined by regions of high deformation rates, which are located along the boundaries of ice floes..."

Page 4 lines 25-26

"radius r1 = 1" and "radius r2 = 5" – please include the UNITS – presumably "pixels" or "grid cells".

Page 5 line 2

"threshold of 15" – what units? Are the pixel values (grid cell values) scaled to the range [0,255] after the histogram equalization, with black=0 and white=255? "threshold of 15" is meaningless unless we know what the scale is.

Page 6 line 15

The sentence "All segments..." references Figure 3(a) but Figure 2 has not been referenced yet. Furthermore, this section (3.1.2) describes segment detection in general, whereas Figure 3(a) is one particular example of segment detection, which is not made clear in the sentence. I'd suggest either deleting this sentence or clarifying that it refers to a particular example.

Page 7 Figure 2

The diagram needs more labeling or the caption needs more explanation. It should be stated explicitly that the red and blue segments are LKFs. What is v? What is a_parallel? What is a_perpendicular? What are the black dotted lines? Etc.

Page 7 Equation (2)

What are alpha and beta? There is no explanation.

Page 8 line 12

What are the units of the deformation rate eps_0 = 0.75? What are the units of the minimum length l_min = 2?

Page 8 end of Section 3.1.3

It might be appropriate to reference Figure 3(a) here, instead of at the end of Section 3.1.2

Page 8 Section 3.1.4

The title of this section is "Parameter optimization", but page 9 lines 9-12 says "finding a global minimum of the cost function is impossible" and "we use a set of parameters estimated with a simple brute-force algorithm" and "We do not regard this set of parameters as the global optimum, but rather as a useful working basis". Therefore this section should not be titled "Parameter optimization" but rather "Parameter selection" or "Parameter determination". There is no optimization taking place, as far as I can tell.

Page 9 Table 1.

The UNITS of the parameters in the table need to be given. I'd suggest adding a column to the table for the units.

Page 11 lines 18-19

While it's true that 18.54% is slightly smaller than 18.74%, it seems to me that the difference is negligible and that these errors are essentially the same.
Page 12 lines 13-14

"The deformation rates in RGPS are known to be prone to grid scale noise" – and more fundamentally, the deformation rates are subject to uncertainty due to tracking and geolocation errors in the underlying SAR images, which have pixel size 100 meters. See:

Lindsay, R.W., and H.L. Stern, 2003, The RADARSAT Geophysical Processor System: Quality of Sea Ice Trajectory and Deformation Estimates, J. Atmos. Ocean. Tech., 20, 1333-1347.

In light of this, is it possible that some of the deforming cells identified in the present paper are actually "in the noise" compared to the background deformation level?

Page 12 line 16

Please give the units of d_LKF = 15

Page 13 line 7

Please give the units of DELTA_eps_0 = 1.25

Page 13 line 8

"common logarithm" and "10ˆ1.25" implies log base 10, but page 4 line 21 says that the "natural logarithm" is used, i.e. log base e. Don't these both refer to log(deformation rate)? If so, then which base is correct, 10 or e? If not, then what is the difference in usage of the log on page 4 vs. the log here on page 13?

Page 14 Figure 7

I can't tell the difference between the points labelled "rounded indices" and the points labelled "1-pixel uncertainty". Perhaps different symbols could be used in the figure.

Page 16 Table 2

The UNITS of the parameters in the table need to be given. I'd suggest adding a

column to the table for the units.

Page 17 Figure 9

It's hard to distinguish the colors of the darker lines. I can see the yellow and green lines, but the other lines all look brown or black to me. Perhaps a different color scheme could be used.

Page 20 lines 21-22

Regarding the distribution of LKF lengths, "The PDF is described accurately by a power-law for LKF length between 60 and 600 km." I strongly dispute that statement. Figure 11(c) shows the PDF. The authors are apparently basing their claim on a least-squares fit in log-log space using evenly spaced bins in LKF length. Such a method leads to a very biased estimate of the power-law exponent; see:

White, EP, Enquist, BJ and Green, JL. 2008. On estimatÂ■ing the exponent of power-law frequency distribuÂ■tions. Ecology 89(4): 905–912. DOI: https://doi.org/10.1890/07-1288.1

and

Clauset, A, Shalizi, CR and Newman, MEJ. 2009. Power-law distributions in empirical data. SIAM Rev 51(4): 661–703. DOI: https://doi.org/10.1137/070710111

The proper way to fit a power law to data is to calculate the Maximum Likelihood Estimate (MLE) of the power-law exponent. After the exponent has been calculated, a goodness-of-fit test should be applied to determine whether or not a power law is in fact an accurate model of the data. It looks to me (Figure 11(c)) as if the PDF has concave-down curvature in log-log space throughout its range. If the authors hypothesize that LKF lengths follow a power-law distribution, they should use the MLE to calculate the exponent, and then apply a goodness-of-fit test to check whether a power-law model does in fact accurately describe the PDF.

Page 21 Figure 11

(i) Figure 11(b) is not described in the text.

(ii) In Figure 11(b) (upper panel), is the y-axis label on the left side of the figure supposed to be "Storm strength" or "Number of cyclones"? The label itself says "Storm stength" [misspelled] but the caption says "number of cyclones".

(iii) In Figure 11(b) (lower panel), the y-axis label on the left side of the figure says "No. of LKFs (normalized)". How is the normalization done? Per unit area? Fraction of the total number of LKFs? The sum of the bars does not seem to be 1.

(iv) In Figure 11(b) (lower panel), what's the story with the gap in early April?

(v) In Figure 11(b) (lower panel) the color (red-yellow) gives the total deformation rate (1/day) from January to May. That time series of deformation rate could be compared with Figure 6 of:

Stern, H. L., and R. W. Lindsay (2009), Spatial scaling of Arctic sea ice deformation, J. Geophys. Res., 114, C10017, doi:10.1029/2009JC005380

in which the total deformation rate (1/day) from January to July is plotted, based on RGPS data.

Page 22 lines 3-4

"exponential distribution with an exponent of 0.34" An exponential distribution can be written in one of two ways: $(1/b)*exp(-t/b)$ where b is a time scale (units: time), or $c*exp(-c*t)$ where c is a rate (units: 1/time).

(i) The authors have not said which form they are using, and therefore the reader does not know whether 0.34 is a time scale or a rate. No units are given.

(ii) In the present case, 0.34 must be a rate with units 1/day. So the time scale is 1/0.34 = 3 days, which makes sense.

[Figure]

(iii) The parameter in an exponential distribution is not called an exponent. It could be called a time scale (first form, b) or a rate (second form, c). In the first form, b is also the mean value. In Figure 11(e), the legend says "exponent 0.34" which is not correct terminology. It could be written "rate 0.34 / day" or "mean value 3 days".

Page 22 lines 13-14 In reference to Figure 11(b), "This overall seasonal cycle is interrupted by a set of four strong cyclones...in March" – I don't see how Figure 11(b) shows four strong cyclones in March. Please tell the reader what to look for in the figure.

=====

Technical Corrections

Page 1 line 14. "is derived" should be "are derived"

Page 2 line 5 and following. Use capital "O" in "Arctic Ocean"

Page 5 line 1 of figure caption. "final output OF the..."

Page 5 line 10. "LKFs" should be "LKF" (singular)

Page 6 line 6. "segments" should be "segment" (singular)

Page 6 line 8. Better to end the sentence after the word "curvature" and start a new sentence: "As in the 2 pixel case, a 90-degree shift..."

Page 7 line 1 of figure caption. "illustration" should be "illustrating"

Page 7 line 3 of figure caption. "lay" should be "lie", and "half-ellipse" should be "half-ellipses" (plural)

Page 7 line 15 and following. Do not abbreviate Table as "Tab." Use "Table"

Page 8 line 15. "The choice of THE set of parameters..."

Page 9 line 19. "to which degree" should be "to what degree"

Page 10 line 3. "The principle idea" should be "The principal idea"

Page 10 line 15. "overlap larger THAN 60%" and "overlap smaller THAN 60%"

Page 10 line 19. "significant" should be "significantly"

Page 12 line 2 of figure caption. "to to" (delete one of them)

Page 12 line 9. Delete the word "also"

Page 12 line 15. "sightly" should be "slightly"

Page 13 line 2. "reconnect" should be "reconnected"

Page 13 line 28. "an" should be "a"

Page 14 line 17. "considered as *a* tracked feature..."

Page 15 line 12. "advecting" should be "advect"

Page 15 line 23. "false positive" should be "false positives" (plural)

Page 15 line 31. "algorithms" should be "algorithm" (singular)

Page 16 line 2 of figure caption. "color" should be "colored", and "shows" should be "show", and "that is" should be "that are"

Page 16 line 5. "where" should be "were"

Page 16 line 6. Delete the comma (,)

Page 18 line 1 of figure caption. "tracks" should be "tracked features"

Page 18 line 2 of figure caption. "tracks" should be "tracked features" and "in growth" should be "into growth"

Page 18 line 4. Delete "for"

Page 18 line 11. Should "in shape" be "as shape"? I'm not sure.

Page 18 line 13. "their" should be "its"

Interactive
comment

Page 19 line 2. "an lifetime" should be "a lifetime"

Page 19 line 7. "where" should be "were"

Page 20 line 2. "winter 2001/01" – should it be "2001/02"?

Page 20 line 24. "47.5 km" should be "37.5 km"

Page 20 line 34. "given in A" should be "given in Appendix A"

Page 21 Figure 11 panel (b). In the title on the vertical axis, "stength" should be "strength". However, the entire title "Storm strength" might be incorrect – should it be "Number of cyclones"?

Page 21 Figure 11 panel (d). The x-axis label should include the units, i.e. "Intersection angle (degrees)"

Page 21 Figure 11 panel (e). The x-axis label should include the units, i.e. "LKF lifetime (days)"

Page 21 line 4 of figure caption. "new" should be "newly"

Page 21 line 6. "less" should be "fewer"

Page 22 line 9. Capitalize "Hemisphere"

Page 22 line 10. "Fig 11" should be "Fig 11(b)"

Page 22 line 11. "general" should be "generally"

Page 22 line 14. "pass the Arctic oceans" should be "pass through the Arctic Ocean"

Page 23 line 17. "11 winters" – I think it's 12 winters: 1996/97 through 2007/08.

Page 23 line 22. "in Arctic ocean" should be "in the Arctic Ocean"

Page 24 line 2. "grid spacing OF 2 km"

Page 24 line 2. Delete "set of"

Page 24 line 5. "for THE entire model domain"

Page 24 line 7. "KOBAYASHI" should be "Kobayashi"

Page 25 lines 23-24. Do not use all capital letters for the names.
* * *

---

## Referee Comment (RC2) · Anonymous Referee #2 · 14 Jan 2019

This paper improved and extended the lead and ridge detection method described in Linow and Dierking 2017, adding a new tracking algorithm. The LKF dataset derived in this study, therefore provides both spatial and temporal statistics of LKFs, which is of great advantage in evaluating sea ice model outputs. The manuscript can be published after minor revision.

General comments:

1. It should be explained in the text why leads and ridges are bounded together to LKFs. If it is impossible to distinguish leads and ridges based on the current ice drift and deformation data, what kind of additional data is needed to get separate information of leads and ridges?

[Figure]

2. The parameter optimization part (section 3.1.4) is not quite convincing. It might be beneficial to separate the hand-picked validation datasets into several randomly selected groups and repeat the parameter optimization procedure, to confirm that the same optimized values can be achieved from different evaluations.

Specific comments:

P1L7: based → based on P2L6: CryosSat-2 → CryoSat2 P3 section 2.1: provide a short description how deformation rate is calculated or give a reference. Also uncertainties in the deformation data should be mentioned. P3 section 2.2: As I understand, sea-ice deformation is calculated from ice drift information, and this same ice drift data is used to track the LKFs? P3L23: visual → visually P4L14: does this deformation rate include shear? It is known that shear is one of main factors to form leads. P6 section 3.1.3 Reconnection: It would nice to illustrate some examples of pairs of segments which could be connected to one LKF and which could not be connected to one LKF according to the three criteria. P8L11: I suppose the unit of D0 and Lmin are pixels? P9L12: Can this strong non-linearity problem be solved if more reference data are available?

---

## Author Comment (AC1) · 30 Jan 2019

**Answers to Anonymous Referee #1**

The authors have taken the sea-ice deformation data from the Radarsat Geophysical Processor System (RGPS) and used it to create a data set of Linear Kinematic Features (LKFs), i.e. leads and ridges that extend across the Arctic pack ice. Spatial coverage includes the western Arctic Ocean with 12.5 km grid size; temporal coverage is 1996-2008 during the winter season (Nov-Apr) with 3-day resolution. The LKFs are tracked over time using a new algorithm. Summary statistics of the LKF data set are presented.

The new data set could prove to be useful for calibration and validation of high- resolution sea-ice models, and as a climatology of LKFs for comparison with a future Arctic sea-ice cover. The paper is mostly a description of algorithms (pages 4-19), but its value lies in the end product – the LKF data set. In my opinion, the paper should be published after the following comments are addressed.

We thank the anonymous referee #1 for his detailed review.

=====

General Comments
Leads are not distinguished from pressure ridges in the data set – they are both labelled as LKFs. Wouldn't it be useful, once an LKF has been identified, to label it as "lead" or "pressure ridge"? Wouldn't it be possible to do this based on the values of divergence within the LKF, with positive divergence indicating a lead and negative divergence indicating a pressure ridge? Leads and pressure ridges are radically different types of features. Leads are the sites of new ice growth, contributing to the thin end of the ice thickness distribution. Pressure ridges affect air and water drag, and contribute to the thick end of the ice thickness distribution. Considering that 165,000 LKFs have been identified, wouldn't it be useful to know what percent of them are leads and what percent are pressure ridges? In my opinion, the authors should either go back and label each LKF as a lead or pressure ridge, or explain why it's impossible or too difficult to do so, or why it doesn't matter.

We fully agree that this topic needs to be discussed in further detail. The idea to use divergence rates to distinguish leads and pressure ridges is obvious, but bears some pitfalls. In general, it is right that leads show divergence and pressure ridges convergence of ice motion and the converse of this relation can be used to label newly formed LKFs. For LKFs with longer lifetime, the converse does not necessarily hold true. Imagine a lead that formed in the previous RGPS image due to diverging ice motion and now starts to close due to convergence. Depending on how strong the converging ice motion is the lead might develop into a pressure ridge or stay a lead: for weak convergence, the lead is just partly closed but still be an opening in the ice cover. For strong convergence, the lead is fully closed and a pressure ridge builds up. Therefore, it is only possible to separate leads and pressure ridges using the divergence data for LKFs for which the sign of the divergence does not change within the lifetime of the LKF. In addition, the spurious noise in divergence rates for RGPS for strongly deformed cells ("Boundary-definition errors" in Lindsay & Stern, 2003, Bouillon & Rampal, 2015) introduces uncertainties in this classification. Combing the LKF data-set with sea-ice thickness and concentration data (that is not provided by RGPS but for model output) allows to clearly distinguish between leads and pressure ridges by using the additional constraints: (1) along a lead the sea-ice concentration decreases within the time step, and (2) along pressure ridges the sea-ice thickness increases. Given these limitations we refrain from classifying all LKFs in the data-set. Nevertheless, we already provide the deformation rates for each LKF in the data-set along with position data to leave the user the option to use this information for classification. An appropriate evaluation of this classification would need to be done by the users themselves. We estimate that the data-set contains 46% leads, 41% pressure ridges, and 13% not-classified LKFs. For this estimate all LKFs are assigned depending on the sign of the divergence rate. If the sign of the divergence changes for LKFs with lifetimes longer 3 days, we only use the divergence data before the sign change for labeling and mark LKF for the remaining lifetime as not-classified. We add a paragraph to Section 5.1 "Generation of LKF data-set" that provides these estimates and a summary of difficulties explained above:

"The deformation, more precisely the divergence rate, which is saved for each LKF, can be used to distinguish leads from pressure ridges in the generation of an LKF. In general, leads form in divergent and pressure ridges in convergent ice motion and the converse of this relation can be used to label newly formed LKFs. Persistent LKFs can also be labeled in this way, as long as the sign of divergence does not change during the lifetime of an LKF. Consider an LKF, initially labelled as a lead in divergence, that encounters convergent motion. Depending on the magnitude of convergence, the lead may either only partly close and continue to be an open lead, or it may close completely and even evolve into a pressure ridge, making differentiating between leads and ridges difficult. Thus, we refrain from labeling all LKFs in the data-set into leads and pressure ridges, but provide the deformation rates for each LKF and leave this classification and its evaluation to the informed user. As an approximate first-guess, we estimate that 46% of the LKFs in the data-set are leads, 41% are pressure ridges, and 13% are unclassified (because the associated mean divergence rate along the LKF changes sign over the lifetime of the LKF). For the classified leads and pressure ridges the sign of divergence does not change over the lifetime. Please note, that these estimates especially for short LKFs are likely contaminated with grid-scale noise in the divergence data of RGPS (Lindsay & Stern, 2003; Bouillon & Rampal, 2015). Combining the LKF data-set with sea-ice thickness and concentration data would allow to clearly distinguish between leads and pressure ridges by using these additional constraints: (1) along a lead the sea-ice concentration decreases within the time step, and (2) along pressure ridges the sea-ice thickness increases."

=====

Specific Comments (in page order)

Page 3 Section 2.1
(i) The RGPS data set needs a reference. Where does it come from? A NASA web site? The Alaska Satellite Facility?
The RGPS is obtained from the Ron Kwok's JPL homepage (https://rkwok.jpl.nasa.gov/radarsat/index.html). We added the link to the manuscript.

(ii) It would be courteous to give credit for the creation of the RGPS, such as:
Kwok R. (1998) The RADARSAT Geophysical Processor System. In: Analysis of SAR Data of the Polar Oceans. Springer, Berlin, Heidelberg. https://doi.org/10.1007/978-3- 642-60282-5_11
We added the suggested reference to the manuscript.

Page 3 Section 2.2
Again the source of the RGPS Lagrangian drift data should be cited.
The RGPS Lagrangian drift data is obtained from the same site as the deformation data. We added the link and reference for the drift data to the manuscript.

Page 3 line 24
Please clarify that "px" = pixel and that 1 pixel = 1 grid cell = 12.5 x 12.5 km.
We clarified this in the manuscript in the following way:
"The intrinsic localization uncertainty of the visually detected features was shown to be 0.75 **pixel** (px) **with 1 pixel corresponding to a grid cell of size 12.5 x 12.5 km** and the…"

Page 4 line 14
After the equation, say that eps_I is divergence and eps_II is shear.
We added this information to the manuscript.

Page 4 lines 15-16
The LKFs are not "marked" by regions of high deformation, they are DEFINED by regions of high deformation. Consider writing: "LKFs are defined by regions of high deformation rates, which are located along the boundaries of ice floes..."

We change the text accordingly.

"radius r1 = 1" and "radius r2 = 5" – please include the UNITS – presumably "pixels" or "grid cells".
Yes, the units are pixels here. We added the units in the remainder of the manuscript accordingly.

"threshold of 15" – what units? Are the pixel values (grid cell values) scaled to the range [0,255] after the histogram equalization, with black=0 and white=255? "threshold of 15" is meaningless unless we know what the scale is.
The threshold describes a difference between two versions of a histogram equalized image that have been smoothed with Gaussian kernels of different size. The values of the image range from 0 (lowest deformation rate) to 255 (highest). Thus it does not have a unit that is directly linked to a length scale or deformation rate. We added the following sentence to the manuscript for clarification:
"The threshold d_LKF does not have an unit as it describes the difference of two histogram equalized images, where the highest deformation rate corresponds to a pixel value 255 and the lowest to value of 0."

The sentence "All segments..." references Figure 3(a) but Figure 2 has not been referenced yet. Furthermore, this section (3.1.2) describes segment detection in general, whereas Figure 3(a) is one particular example of segment detection, which is not made clear in the sentence. I'd suggest either deleting this sentence or clarifying that it refers to a particular example.
We changed the order of the figures and clarified that Figure 3(a) refers to the particular example of January 1st, 2006.

The diagram needs more labeling or the caption needs more explanation. It should be stated explicitly that the red and blue segments are LKFs. What is v? What is a_parallel? What is a_perpendicular? What are the black dotted lines? Etc.
We label the two segments with A and B and explain all variables in the figure in the caption. The modified caption reads as follows:
"Sketch of two segments **A and B** illustrating the principle behind the elliptical distance. In the distance computation the component pointing in the direction perpendicular to the segment is weighted by a factor of e. Within the shaded area the elliptical distance is below a threshold D_0. If the endpoints of both segments lie within the area where both shaded half-ellipse overlap, they are considered for reconnection. **The dotted lines indicate the orientation of lines connecting the start and end points of both segments and the angle ΔO is the difference in orientation. v is the vector connecting the endpoints of both segments as defined in Eq.(2). a_parallel and a_perpendicular are basis vectors aligned in the direction of the segment A, respectively b_parallel and b_perpendicular for B.**"

What are alpha and beta? There is no explanation.
alpha and beta are the coefficients of the vector decomposition into the different basis (a_parallel, a_perpendicular) and (b_parallel, b_perpendicular). We added this explanation to the description of Equation (2).

What are the units of the deformation rate eps_0 = 0.75? What are the units of the minimum length l_min = 2?
The minimum length l_min is given in pixels. The deformation rates eps_0 is log(1/day), because it is applied to the common logarithm of deformation fields with the unit 1/day.

It might be appropriate to reference Figure 3(a) here, instead of at the end of Section 3.1.2
We changed the order of the figures and the references, please see answer above.

Page 8 Section 3.1.4
The title of this section is "Parameter optimization", but page 9 lines 9-12 says "finding a global minimum of the cost function is impossible" and "we use a set of parameters estimated with a simple brute-force algorithm" and "We do not regard this set of parameters as the global optimum, but rather as a useful working basis". Therefore this section should not be titled "Parameter optimization" but rather "Parameter selection" or "Parameter determination". There is no optimization taking place, as far as I can tell.
We agree that we do not perform an optimization in a strict mathematical sense, because we choose the parameters that minimize the number of not-detected features from $5^6=15625$ different sets of parameters. We renamed the section title to "Parameter selection".

Page 9 Table 1.
The UNITS of the parameters in the table need to be given. I'd suggest adding a column to the table for the units.
We added a column with all units to Table 1.

Page 11 lines 18-19
While it's true that 18.54% is slightly smaller than 18.74%, it seems to me that the difference is negligible and that these errors are essentially the same.
We agree and rephrased the sentence to:
"For both algorithms, the distributions are similar with **similar** mean errors of ~18%."

Page 12 lines 13-14
"The deformation rates in RGPS are known to be prone to grid scale noise" – and more fundamentally, the deformation rates are subject to uncertainty due to tracking and geolocation errors in the underlying SAR images, which have pixel size 100 meters. See:
Lindsay, R.W., and H.L. Stern, 2003, The RADARSAT Geophysical Processor System: Quality of Sea Ice Trajectory and Deformation Estimates, J. Atmos. Ocean. Tech., 20, 1333-1347.
We agree and added the reference to the text.
In light of this, is it possible that some of the deforming cells identified in the present paper are actually "in the noise" compared to the background deformation level?
We implemented two instances in the algorithm to prevent that features are detected due to their high noise level: (1) we use a threshold for the DoG filtered deformation field that is d_LKF=15. This guarantees that only pixels are marked as LKF if the local deformation rate is much higher than the local surrounding, which suppresses small scale noise. (2) Detected features need to have a minimum length of l_min=3px. So, for falsely detecting an LKF due to noise, a minimum of three neighbouring LKFs-marked cells are needed with comparable noise-induced high deformation rates. This case is unlikely to happen.

Page 12 line 16
Please give the units of d_LKF = 15
All units are added to Table 1 and added in the text.

Page 13 line 7
Please give the units of DELTA_eps_0 = 1.25 Page 13 line 8
"common logarithm" and "10^1.25" implies log base 10, but page 4 line 21 says that the "natural logarithm" is used, i.e. log base e. Don't these both refer to log(deformation rate)? If so, then which base is correct, 10 or e? If not, then what is the difference in usage of the log on page 4 vs. the log here on page 13?
In the "Data preprocessing and filtering" section we apply the natural logarithm (shown in Fig.1b), i.e. log base e, as it was done in Linow & Dierking, 2017. In the segment reconnection, however, the threshold Δeps_0 is applied to the

common logarithm of the total deformation, i.e. log base 10, so that Δeps_0 directly describes the difference in order of magnitude. Thus, we use logarithms with different bases on purpose and compute those in both cases from the original input total deformation fields. In the end, one could also use the common logarithm in the filtering without having any difference in the detected LKFs, because next the logarithmic deformation fields are histogram equalized. As logarithmic fields can be converted to a different base by multiplying with a constant, the histogram equalized version of two logarithmic fields with different base are identical.

We added further information to the definition of Δeps_0 to clarify that we are using logarithms with different bases and explain why:

"… and the difference of the logarithm of the total deformation rate Δlog_10(eps_0). We here use the common logarithm, i.e. log base 10, in contrast to the natural logarithm used in Section 3.1.1 to directly describe the difference in the order of magnitude in the total deformation of two segments."

Page 14 Figure 7
I can't tell the difference between the points labelled "rounded indices" and the points labelled "1-pixel uncertainty". Perhaps different symbols could be used in the figure.
We changed the symbols in Figure 7 to filled and not-filled circles that are easy to distinguish.

Page 16 Table 2
The UNITS of the parameters in the table need to be given. I'd suggest adding a C5 column to the table for the units.
We added a new column to the table that provides the units of all parameters.

Page 17 Figure 9
It's hard to distinguish the colors of the darker lines. I can see the yellow and green lines, but the other lines all look brown or black to me. Perhaps a different color scheme could be used.
We changed the colormap of the plot:

[Figure]

Page 20 lines 21-22
Regarding the distribution of LKF lengths, "The PDF is described accurately by a power-law for LKF length between 60 and 600 km." I strongly dispute that statement. Figure 11(c) shows the PDF. The authors are apparently basing their claim on a least- squares fit in log-log space using evenly spaced bins in LKF length. Such a method leads to a very biased estimate of the power-law exponent; see:

White, EP, Enquist, BJ and Green, JL. 2008. On estimate ing the exponent of power-law frequency distributions. Ecology 89(4): 905–912. DOI: https://doi. org/10.1890/07-1288.1
and
Clauset, A, Shalizi, CR and Newman, MEJ. 2009. Power-law distributions in empirical data. SIAM Rev 51(4): 661–703. DOI: https://doi.org/10.1137/070710111

The proper way to fit a power law to data is to calculate the Maximum Likelihood Estimate (MLE) of the power-law exponent. After the exponent has been calculated, a goodness-of-fit test should be applied to determine whether or not a power law is in fact an accurate model of the data. It looks to me (Figure 11(c)) as if the PDF has concave-down curvature in log-log space throughout its range. If the authors hypothesize that LKF lengths follow a power-law distribution, they should use the MLE to calculate the exponent, and then apply a goodness-of-fit test to check whether a power-law model does in fact accurately describe the PDF.

Thanks for catching this. We performed a new fit of a power law to the data using a Maximum Likelihood Estimator. We use the methodology described in Stern et al., 2018a in the context of floe size distributions, which is based on White et al., 2008 and Clauset et al., 2009. The goodness-of-fit test follows Stern et al., 2018b and is a bootstrap method of the Kolmogorov-Smirnov (KS) statistic evaluated for 1000 data sets that were randomly drawn from the fitted power-law distribution. The KS is the distance between the cumulative distributions function (CDF) of the data and the model. The KS is computed for the observed data as well as for all 1000 random data-sets that have the same number of samples. If the KS of the observed data is greater than the 95% percentile of the KS of the random data-sets, the observed data is not described by a power-law distribution. With this method, we find that the LKF length does not follow a power-law distribution over the whole range. Revisiting these calculations, we find that a stretched exponential distribution ( p(x) = C * x^(beta-1) *exp(-lambda*x^beta) , Clauset et al., 2009) describes the observed LKF length better. We determined the both parameters lambda and beta by maximizing the log-likelihood. The goodness-of-fit test described above finds that LKF length follows a stretched exponential distribution for in the range 100km to 1000km. Smaller scales needed to be neglected in the goodness-of-fit test because of the perturbations in the CDF due to the discrete length scales. As LKFs are described as a collection of pixels, their length is limited to (i+j*sqrt(2)) * resolution with i,j being integer values. The plot is changed accordingly and the text describing it reads as follows:

"The distributions of LKF length of all single years are very similar and range from 50 km to 1000 km (Fig. 11c). For LKF lengths between 100 and 1000 km, a stretched exponential distribution, p(x) = C*x^(beta-1)*e^(-lambda*x^beta) with C=beta*lambda*e^(lambda*x_min^beta) (Clauset et al.,2009), accurately describes the PDF.

The parameters beta=0.719 and lambda=0.0531 are determined by finding numerically the maximum likelihood estimate. We perform a goodness-of-fit test for power-law distributed data that is based on the Kolmogorov-Smirnov (KS) statistic, which is the maximum distance between the cumulative distribution functions (CDF) of two different distributions (Clauset et al., 2009). We draw random samples from the fitted distribution and compute the KS of the random samples and the fitted distribution. This simulation is repeated 1000 times. We find that the KS of the observed length scales is smaller than the 95% percentile of the random samples and thereby the observed LKF lengths are described by the fitted distribution.

The stretched exponential distribution belongs to the family of heavy-tailed distributions, and is the transition of an exponential and a power-law distribution. It describes many natural phenomena that are dominated by extreme events but in contrast to power-law distribution have a natural upper limit scale (Laherrere & Sornette, 1998).

We limit the range of the fit to 1000 km due to the varying coverage and especially smaller gaps in the RGPS data that may divide long features into multiple smaller segments. We set the lower bound to 100km to account for the discrete character of the LKF length that disturbs the distribution at lower LKF lengths. As LKFs here are collections of pixels, their length is set by a linear combination of (i+j*sqrt(2))*12.5km with i,j element N."

Page 21 Figure 11 (b)
(i) Figure 11(b) is not described in the text.
This was due to a wrong reference in the text that was corrected.

(ii) In Figure 11(b) (upper panel), is the y-axis label on the left side of the figure supposed to be "Storm strength" or "Number of cyclones"? The label itself says "Storm stength" [misspelled] but the caption says "number of cyclones". This was a typo and is now corrected.

(iii) In Figure 11(b) (lower panel), the y-axis label on the left side of the figure says "No. of LKFs (normalized)". How is the normalization done? Per unit area? Fraction of the total number of LKFs? The sum of the bars does not seem to be 1.

The number of LKFs is normalized with the overall coverage of the RGPS time step. Thus the "No. of LKFs (normalized)" provides the fraction of the RGPS cells that are LKFs showing a certain deformation (color-code). We changed the caption accordingly.

(iv) In Figure 11(b) (lower panel), what's the story with the gap in early April?
For this date no regular RGPS data is available, and therefore no LKF data. We include this note in the text.

(v) In Figure 11(b) (lower panel) the color (red-yellow) gives the total deformation rate (1/day) from January to May. That time series of deformation rate could be compared with Figure 6 of:
Stern, H. L., and R. W. Lindsay (2009), Spatial scaling of Arctic sea ice deformation, J. Geophys. Res., 114, C10017, doi:10.1029/2009JC005380
in which the total deformation rate (1/day) from January to July is plotted, based on RGPS data.

We also commented now on the overall seasonal cycle of the deformation associated with each LKF and reference the mentioned paper:

"The deformation rates associated with the LKFs co-varies with the seasonal cycle of the number of LKFs, which is in agreement with the seasonal cycle of the mean deformation rate (Stern & Lindsay, 2009)"

Page 22 lines 13-14
In reference to Figure 11(b), "This overall seasonal cycle is interrupted by a set of four strong cyclones...in March" – I don't see how Figure 11(b) shows four strong cyclones in March. Please tell the reader what to look for in the figure.
We clarified the caption of Figure 11(b) to make clear what is shown in the two panels:

"(b) Time series of the number of LKFs separated by their deformation rates for winter 2002/03. The number of LKFs is normalized by the number of available pixels for each RGPS time records. For early April no RGPS data is available. For each time record all cyclones (Serreze et al., 2009) over the Arctic ocean are each shown as a dot in the upper panel. The y-axis and the color-code of the dots in the upper panel provide the local Laplacian of each cyclone as a proxy for its strength."

Page 22 lines 3-4
"exponential distribution with an exponent of 0.34" An exponential distribution can be written in one of two ways: (1/b)*exp(-t/b) where b is a time scale (units: time), or c*exp(-c*t) where c is a rate (units: 1/time).
(i) The authors have not said which form they are using, and therefore the reader does not know whether 0.34 is a time scale or a rate. No units are given.
(ii) In the present case, 0.34 must be a rate with units 1/day. So the time scale is 1/0.34 = 3 days, which makes sense.
(iii) The parameter in an exponential distribution is not called an exponent. It could be called a time scale (first form, b) or a rate (second form, c). In the first form, b is also the mean value. In Figure 11(e), the legend says "exponent 0.34" which is not correct terminology. It could be written "rate 0.34 / day" or "mean value 3 days".
Being more precise, the distribution of LKF lifetimes shows an exponential tail of rate 0.34 with the unit 1/day. We determine the rate of the tail by an least square fit of the PDF to the lifetime, as suggested in Sornette (2006) to get a first estimate of the rate. In doing so, we do not determine the rate of a exponential distribution in a strict sense. Using a Maximum Likelihood Estimator to determine the rate of a exponential distribution shows that the LKF lifetime does not follow an exponential distribution due to the overestimation of short-lived LKFs (lifetime of 0 to 3 days). We hypothesize that this overestimation originates from the changing coverage of RGPS that prevents some LKFs to be tracked. Therefore, we provide the rate of the exponential tail and changed the wording of the manuscript accordingly: we speak of exponential tail instead of distribution, and rate instead of exponent. Also the units are now provided.

=====

Technical Corrections

We addressed all following corrections. Thank you for catching them!

Page 1 line 14. "is derived" should be "are derived"
Page 2 line 5 and following. Use capital "O" in "Arctic Ocean"
Page 5 line 1 of figure caption. "final output OF the..."
Page 5 line 10. "LKFs" should be "LKF" (singular)
Page 6 line 6. "segments" should be "segment" (singular)
Page 6 line 8. Better to end the sentence after the word "curvature" and start a new sentence: "As in the 2 pixel case, a 90-degree shift..."
Page 7 line 1 of figure caption. "illustration" should be "illustrating"
Page 7 line 3 of figure caption. "lay" should be "lie", and "half-ellipse" should be "half- ellipses" (plural)
Page 7 line 15 and following. Do not abbreviate Table as "Tab." Use "Table"
Page 8 line 15. "The choice of THE set of parameters..."
Page 9 line 19. "to which degree" should be "to what degree"
Page 10 line 3. "The principle idea" should be "The principal idea" C8
Page 10 line 15. "overlap larger THAN 60%" and "overlap smaller THAN 60%"
Page 10 line 19. "significant" should be "significantly"
Page 12 line 2 of figure caption. "to to" (delete one of them)
Page 12 line 9. Delete the word "also"
Page 12 line 15. "sightly" should be "slightly"
Page 13 line 2. "reconnect" should be "reconnected"
Page 13 line 28. "an" should be "a"
Page 14 line 17. "considered as *a* tracked feature..."
Page 15 line 12. "advecting" should be "advect"
Page 15 line 23. "false positive" should be "false positives" (plural)
Page 15 line 31. "algorithms" should be "algorithm" (singular)
Page 16 line 2 of figure caption. "color" should be "colored", and "shows" should be "show", and "that is" should be "that are"
Page 16 line 5. "where" should be "were"
Page 16 line 6. Delete the comma (,)
Page 18 line 1 of figure caption. "tracks" should be "tracked features"
Page 18 line 2 of figure caption. "tracks" should be "tracked features" and "in growth" should be "into growth"
Page 18 line 4. Delete "for"
Page 18 line 11. Should "in shape" be "as shape"? I'm not sure.
Page 18 line 13. "their" should be "its"
Page 19 line 2. "an lifetime" should be "a lifetime"
Page 19 line 7. "where" should be "were"
Page 20 line 2. "winter 2001/01" – should it be "2001/02"?
Page 20 line 24. "47.5 km" should be "37.5 km"
Page 20 line 34. "given in A" should be "given in Appendix A"
Page 21 Figure 11 panel (b). In the title on the vertical axis, "stength" should be "strength". However, the entire title "Storm strength" might be incorrect – should it be "Number of cyclones"?
Page 21 Figure 11 panel (d). The x-axis label should include the units, i.e. "Intersection angle (degrees)"

Page 21 Figure 11 panel (e). The x-axis label should include the units, i.e. "LKF lifetime (days)"

Page 21 line 4 of figure caption. "new" should be "newly"

Page 21 line 6. "less" should be "fewer"

Page 22 line 9. Capitalize "Hemisphere"

Page 22 line 10. "Fig 11" should be "Fig 11(b)"

Page 22 line 11. "general" should be "generally"

Page 22 line 14. "pass the Arctic oceans" should be "pass through the Arctic Ocean"

Page 23 line 17. "11 winters" – I think it's 12 winters: 1996/97 through 2007/08.

Page 23 line 22. "in Arctic ocean" should be "in the Arctic Ocean"

Page 24 line 2. "grid spacing OF 2 km"

Page 24 line 2. Delete "set of"

Page 24 line 5. "for THE entire model domain"

Page 24 line 7. "KOBAYASHI" should be "Kobayashi"

Page 25 lines 23-24. Do not use all capital letters for the names.

---

## Author Comment (AC2) · 30 Jan 2019

Answers to Anonymous Referee #2

This paper improved and extended the lead and ridge detection method described in Linow and Dierking 2017, adding a new tracking algorithm. The LKF dataset derived in this study, therefore provides both spatial and temporal statistics of LKFs, which is of great advantage in evaluating sea ice model outputs. The manuscript can be published after minor revision.

We thank the anonymous referee #2 for his review.

========

General comments:
1. It should be explained in the text why leads and ridges are bounded together to LKFs. If it is impossible to distinguish leads and ridges based on the current ice drift and deformation data, what kind of additional data is needed to get separate information of leads and ridges?

*This answer is the same as for referee #1 who had the same comment.*

We fully agree that this topic needs to be discussed in further detail. The idea to use divergence rates to distinguish leads and pressure ridges is obvious, but bears some pitfalls. In general, it is right that leads show divergence and pressure ridges convergence of ice motion and the converse of this relation can be used to label newly formed LKFs. For LKFs with longer lifetime, the converse does not necessarily hold true. Imagine a lead that formed in the previous RGPS image due to diverging ice motion and now starts to close due to convergence. Depending on how strong the converging ice motion is the lead might develop into a pressure ridge or stay a lead: for weak convergence, the lead is just partly closed but still be an opening in the ice cover. For strong convergence, the lead is fully closed and a pressure ridge builds up. Therefore, it is only possible to separate leads and pressure ridges using the divergence data for LKFs for which the sign of the divergence does not change within the lifetime of the LKF. In addition, the spurious noise in divergence rates for RGPS for strongly deformed cells ("Boundary-definition errors" in Lindsay & Stern, 2003, Bouillon & Rampal, 2015) introduces uncertainties in this classification. Combing the LKF data-set with sea-ice thickness and concentration data (that is not provided by RGPS but for model output) allows to clearly distinguish between leads and pressure ridges by using the additional constraints: (1) along a lead the sea-ice concentration decreases within the time step, and (2) along pressure ridges the sea-ice thickness increases. Given these limitations we refrain from classifying all LKFs in the data-set. Nevertheless, we already provide the deformation rates for each LKF in the data-set along with position data to leave the user the option to use this information for classification. An appropriate evaluation of this classification would need to be done by the users themselves. We estimate that the data-set contains 46% leads, 41% pressure ridges, and 13% not-classified LKFs. For this estimate all LKFs are assigned depending on the sign of the divergence rate. If the sign of the divergence changes for LKFs with lifetimes longer 3 days, we only use the divergence data before the sign change for labeling and mark LKF for the remaining lifetime as not-classified. We add a paragraph to Section 5.1 "Generation of LKF data-set" that provides these estimates and a summary of difficulties explained above:
"The deformation, more precisely the divergence rate, which is saved for each LKF, can be used to distinguish leads from pressure ridges in the generation of an LKF. In general, leads form in divergent and pressure ridges in convergent ice motion and the converse of this relation can be used to label newly formed LKFs. Persistent LKFs can also be labeled in this way, as long as the sign of divergence does not change during the lifetime of an LKF. Consider an LKF, initially labelled as a lead in divergence, that encounters convergent motion. Depending on the magnitude of convergence, the lead may either only partly close and continue to be an open lead, or it may close completely and even evolve into a pressure ridge, making differentiating between leads and ridges difficult. Thus, we refrain from labeling all LKFs in the data-set into leads and pressure ridges, but provide the deformation rates for each LKF and leave this classification and its evaluation to the informed user. As an approximate first-guess, we estimate that 46% of the LKFs in the data-set are leads, 41% are pressure ridges, and 13% are unclassified (because the associated

mean divergence rate along the LKF changes sign over the lifetime of the LKF). For the classified leads and pressure ridges the sign of divergence does not change over the lifetime. Please note, that these estimates especially for short LKFs are likely contaminated with grid-scale noise in the divergence data of RGPS (Lindsay & Stern, 2003; Bouillon & Rampal, 2015). Combining the LKF data-set with sea-ice thickness and concentration data would allow to clearly distinguish between leads and pressure ridges by using these additional constraints: (1) along a lead the sea-ice concentration decreases within the time step, and (2) along pressure ridges the sea-ice thickness increases."

2. The parameter optimization part (section 3.1.4) is not quite convincing. It might be beneficial to separate the hand-picked validation datasets into several randomly selected groups and repeat the parameter optimization procedure, to confirm that the same optimized values can be achieved from different evaluations.
We agree that we do not perform an optimization in a strict mathematical sense, because we choose the parameters that minimize the number of not-detected features from 5^6=15625 different sets of parameters. Therefore, we renamed the section title to "Parameter selection". Nevertheless, we tested the robustness of our "optimal" set of parameters by repeating the parameter selection process for five random subsamples of the evaluation data set as suggested. For each subsample we randomly pick half of the features in the evaluation data-set. For two of those random subsamples the "optimal" set of parameters that was determined using all evaluation data also minimise the number of not-detected features of the subsampled features. For the other random subsamples we find 1, 2, and 44 set of parameters showing a 0.24%, 0.48%, and 2.39% higher amount of detected features. The stronger deviations of one subsample can be explained by the too small amount of long LKFs by random subsampling given that the distribution of LKF length is heavy tailed. Because the deviations of different subsets are small, we conclude that the "optimal" set of parameters is the best estimate obtained for the limited amount of evaluation data. For a more robust optimization that aims to find a global minimum of the number of not-detected features would require a larger evaluation data set. We attempt to make this clear by our extensive discussion of this problem in the manuscript and by the renaming of the section.

========

Specific comments:

P1L7: based → based on
We changed the manuscript accordingly.

P2L6: CryosSat-2 → CryoSat2
We changed the manuscript accordingly.

P3 section 2.1: provide a short description how deformation rate is calculated or give a reference. Also uncertainties in the deformation data should be mentioned.
We now cite Lindsay & Stern (2003) in the description of the deformation data to refer the interested reader to the equations of the line integral approximation.

P3 section 2.2: As I understand, sea-ice deformation is calculated from ice drift information, and this same ice drift data is used to track the LKFs?
In principle, yes. RGPS, however, provides drift information only as a Lagrangian data-set, whereas deformation is also available interpolated to an Eulerian grid. As the tracking algorithm works on regular meshes, we interpolate the Lagrangian drift data-set to the same Eulerian grid used for the deformation data-set.

P3L23: visual → visually
We changed the manuscript accordingly.

P4L14: does this deformation rate include shear? It is known that shear is one of main factors to form leads.

The total deformation rate that the algorithm uses is sqrt(divergence**2 + shear**2). Thus, it includes shear. We clarified this in the manuscript:

"The standard input data of the LKF detection algorithm is the total deformation rate of sea ice epsilon_tot=sqrt(epsilon_I^2+epsilon_II^2), where epsilon_I is the divergence and epsilon_II the shear both of which can be derived from both satellite data and model output."

P6 section 3.1.3 Reconnection: It would be nice to illustrate some examples of pairs of segments which could be connected to one LKF and which could not be connected to one LKF according to the three criteria.

We assume you had a figure similar to Fig. 5 in Linow & Dierking (2017) in mind that illustrates the simple(r) reconnection scheme of the original algorithm (minimising the difference in orientation for all segments that have elliptical distance below a certain threshold). In our case, the reconnection scheme is more complex by taking into account also the difference in deformation rate and by weighting the three contributions of the probability estimate (difference in elliptical distance, in orientation, and in deformation rate). The elliptical difference and difference of orientation are already illustrated in Figure 2 of the manuscript. We can not think of a more compact way of illustrating the additional weighting of the three terms than the mathematical description given in Eq. (1). As the manuscript already includes 11 figures, we refrain from adding a new figure because we are afraid that it would add more complexity and would not facilitate the understanding.

P8L11: I suppose the unit of D0 and Lmin are pixels?

Yes, both are given in pixels. Following the suggestions of reviewer #1 we extended Table 1 and 2 with a column including the units of all parameters used.

P9L12: Can this strong non-linearity problem be solved if more reference data are available?

A larger amount of reference data would dampen the non-linearity in the cost function. This is already explained in the conclusions of the manuscript (P22 L31-32). However, creating such a reference data set by hand-picking is time consuming and beyond the scope of this paper.